# T cell receptor signaling strength establishes the chemotactic properties of effector CD8+ T cells that control tissue-residency

Mahmoud Abdelbary[1], Samuel J. Hobbs[1], James S. Gibbs[2], Jonathan W. Yewdell [2] & Jeffrey C. Nolz [1,3,4] ✉

Tissue-resident memory ($T_{RM}$) CD8+ T cells are largely derived from recently activated effector T cells, but the mechanisms that control the extent of $T_{RM}$ differentiation within tissue microenvironments remain unresolved. Here, using an IFNγ-YFP reporter system to identify CD8+ T cells executing antigen-dependent effector functions, we define the transcriptional consequences and functional mechanisms controlled by TCR-signaling strength that occur within the skin during viral infection to promote $T_{RM}$ differentiation. TCR-signaling both enhances CXCR6-mediated migration and suppresses migration toward sphingosine-1-phosphate, indicating the programming of a 'chemotactic switch' following secondary antigen encounter within non-lymphoid tissues. Blimp1 was identified as the critical target of TCR re-stimulation that is necessary to establish this chemotactic switch and for $T_{RM}$ differentiation to efficiently occur. Collectively, our findings show that access to antigen presentation and strength of TCR-signaling required for Blimp1 expression establishes the chemotactic properties of effector CD8+ T cells to promote residency within non-lymphoid tissues.

Following their activation and proliferative expansion in response to infection or vaccination, effector CD8+ T cells enter the circulation and subsequently infiltrate inflamed, nonlymphoid tissues[1]. Some of these recently activated effector T cells become permanently retained within nonlymphoid tissue microenvironments and differentiate into long-lived tissue-resident memory ($T_{RM}$) CD8+ T cells[2,3]. Due to their anatomical location, $T_{RM}$ CD8+ T cells are poised to rapidly execute effector functions and initiate inflammatory responses following pathogen detection[4–7]. This potent inflammatory capacity can also be detrimental and result in immunopathology, as $T_{RM}$ T cells have also been suggested to be the causative agent of several autoinflammatory diseases including psoriasis, allergic contact dermatitis, and inflammatory bowel disease[8]. Accordingly, a more complete understanding of the mechanisms that either promote or limit the formation of this

important cell type could be instructive for advancing rational vaccine design and immunotherapy approaches, but also potentially for the treatment of autoinflammatory conditions that occur within non-lymphoid tissues such as the skin.

The differentiation of $T_{RM}$ CD8+ T cells is accompanied by the engagement of unique transcriptional programs that enforce permanent residence within tissue microenvironments. Several transcription factors have been implicated in controlling $T_{RM}$ differentiation including Runx3, Notch, Nr4a1, Hobit, and Blimp1[9–12]. It is believed that these transcription factors govern $T_{RM}$ differentiation, at least in part, by both promoting tissue-retention and suppressing gene networks that would support the development of circulating memory T cells. $T_{RM}$ CD8+ T cells formed within distinct nonlymphoid tissues also express a unique repertoire of chemokine receptors including CCR8,

[1]Department of Molecular Microbiology and Immunology, Oregon Health & Science University, Portland, OR, USA. [2]Cellular Biology Section, Laboratory of Viral Diseases, National Institute of Allergy and Infectious Diseases, National Institutes of Health, Bethesda, MD, USA. [3]Department of Cell, Developmental and Cancer Biology, Oregon Health & Science University, Portland, OR, USA. [4]Department of Dermatology, Oregon Health & Science University, Portland, OR, USA. ✉e-mail: nolz@ohsu.edu

CCR9, CCR10 and CXCR6[13–15], but the individual requirement for specific chemokine receptors in $T_{RM}$ differentiation remains ill-defined. CXCR6 and CCR8 are both expressed by skin $T_{RM}$ CD8[+] T cells, however, CXCR6 was found to be of particular importance for the maintenance of skin $T_{RM}$ as genetic deletion of CXCR6, but not CCR8, significantly decreased the number of $T_{RM}$ that form within the epidermis[13]. Moreover, $T_{RM}$ CD8[+] T cells within other nonlymphoid compartments consistently express high levels of CXCR6 and is also critical for the formation and/or maintenance of $T_{RM}$ T cells within the liver and lung without affecting the trafficking of recently activated effector CD8[+] T cells into those tissues[16,17]. While there is clearly the potential for multiple signaling pathways and chemokine receptors to be engaged by recently activated CD8[+] T cells that enforce their residence, how the integration of diverse signaling pathways within tissue microenvironments collectively control the development of $T_{RM}$ T cells in vivo is not completely understood.

Within inflamed nonlymphoid tissues, effector CD8[+] T cells are exposed to a range of tissue-derived factors that regulate particular aspects of the $T_{RM}$ transcriptional program. For example, inflammatory cytokines such as TNF, IL-33, or type I IFNs contribute to repression of $S1pr1$[18], and act synergistically with TGF-β, a well-characterized driver of CD103 expression, suggesting that the local cytokine environment regulates the acquisition of the $T_{RM}$ phenotype. TGF-β responsiveness is generally required for $T_{RM}$ development and entry into the tissue microenvironment can be sufficient to promote $T_{RM}$ differentiation[14,19,20]. Persistent antigen presentation in the small intestine inhibits CD103 expression, suggesting that continuous antigen recognition can prevent TGF-β-mediated $T_{RM}$ differentiation[21]. In contrast, antigen recognition has been shown to be necessary for optimal $T_{RM}$ formation following viral infection of the brain or lung[22,23]. Finally, access to cognate antigen strongly enhances $T_{RM}$ CD8[+] T cell retention and differentiation during acute Vaccinia virus (VacV) infection of the skin[24,25]. Thus, whether antigen recognition within nonlymphoid tissues ultimately promotes or limits $T_{RM}$ differentiation remains controversial.

Here, we use an IFNγ-YFP reporter system to definitively identify effector CD8[+] T cells actively receiving TCR stimulation within the skin microenvironment during VacV infection. The fraction of effector CD8[+] T cells receiving antigenic stimulation and executing effector functions in the skin exhibit a gene transcription profile indicating $T_{RM}$ differentiation, whereas T cells not producing IFNγ resemble precursors of circulating memory T cells. Mechanistically, we show that strength of TCR stimulation causes effector CD8[+] T cells to alter their chemotactic properties that control tissue residency, where low affinity agonists are sufficient to promote CXCR6-mediated migration, but strong TCR signaling is required to prevent S1P-mediated egress. Finally, we find that effector, but not naïve CD8[+] T cells, rapidly upregulate expression of the transcription factor Blimp1 only following secondary antigen encounter, which is required for both the switch in chemotactic properties and for $T_{RM}$ differentiation to occur. Overall, our findings suggest that effector CD8[+] T cells actively engaging in TCR-dependent effector functions within nonlymphoid tissues are major $T_{RM}$ precursors, findings which could provide insights into therapies designed to either enhance or limit the formation of $T_{RM}$ CD8[+] T cells within nonlymphoid tissues such as the skin.

## Results

### Only a fraction of tissue-infiltrating effector CD8[+] T cells express IFNγ in an antigen-dependent manner during viral skin infection

Epicutaneous infection with VacV generates robust populations of $T_{RM}$ CD8[+] T cells in a manner that is highly dependent on local recognition of cognate antigen within the skin microenvironment[24–26]. To understand the mechanisms by which local antigen encounter promotes $T_{RM}$ CD8[+] T cell differentiation, we sought to identify the effector CD8[+] T cells that were actively receiving TCR stimulation within VacV-

infected skin, as it is unclear whether all antigen-specific T cells engage in cognate antigen recognition or if spatiotemporal factors within the tissue microenvironment limits their access to antigen-presenting cells. To do this, we utilized TCR-transgenic (TCR-tg) P14 CD8[+] T cells (specific for the LCMV-derived epitope GP$_{33-41}$ presented by H-2D[b]) that expressed a single copy of an IFNγ-YFP reporter gene. To evaluate whether the IFNγ-YFP reporter system accurately reflected IFNγ protein expression, naïve IFNγ-YFP P14 CD8[+] T cells were transferred into naïve B6 mice that were then infected with VacV expressing GP$_{33-41}$ (VacV-GP33) on the left ear skin. On day 7 postinfection, we stimulated P14 CD8[+] T cells from the spleen with increasing concentrations of GP$_{33-41}$ and directly compared YFP expression to intracellular IFNγ staining. The percentage of T cells expressing YFP and IFNγ protein was equivalent at all peptide concentrations and the kinetics of YFP decay after peptide was removed indicated a YFP half-life of ~14.2 hours. (Supplementary Fig. 1a–c). IFNγ protein expression (directly ex vivo) was found only in YFP[+] P14 CD8[+] T cells in the skin during VacV-GP33 infection (Supplementary Fig. 1d, e). Collectively, these data demonstrate that YFP faithfully reports expression of IFNγ in an antigen-dependent manner.

Having established the utility and functional characteristics of the IFNγ-YFP reporter, we next investigated the spatial and temporal expression of IFNγ by antigen-specific CD8[+] T cells during the course of a viral skin infection. IFNγ expression was essentially undetectable in T cells isolated from lymphoid organs, but ~20-30% of P14 CD8[+] T cells within VacV-infected skin consistently expressed IFNγ during days 4–7 postinfection (Fig. 1a–c and Supplementary Fig. 2a). IFNγ expression by P14 CD8[+] T cells was lost coincident with viral clearance, which occurs ~10–15 days postinfection[24]. Essentially all IFNγ-YFP expressing P14 CD8[+] T cells were protected from intravenous labeling used to identify the T cells associated with the vasculature and YFP[+] T cells were also slightly enriched in the epidermis compared with the dermis (Fig. 1d–g). Exposure to particular combinations of inflammatory cytokines can be sufficient to cause effector CD8[+] T cells to express IFNγ[27]. To test if IFNγ expression by CD8[+] T cells in VacV-infected skin required recognition of cognate antigen, we co-infected mice on the right ear skin with VacV (-Ag) and on the left ear skin with VacV-GP33 (+Ag) (Fig. 1h). Recruitment of effector CD8[+] T cells into the skin is inflammation-dependent but antigen-independent[24,28,29], and thus, effector P14 CD8[+] T cells were recruited equally to both sites of infection (Fig. 1i, j), as similar local inflammatory environments were caused by both VacV infections. However, IFNγ expression was highly enriched within the +Ag skin compared with -Ag skin lacking expression of the immunogenic peptide (Fig. 1k), Taken together, these data demonstrate that infiltration into antigen-rich, VacV-infected skin is required for antigen-specific CD8[+] T cells to express IFNγ.

Effector CD8[+] T cells that infiltrated VacV-infected skin were predominately KLRG1[-] compared with T cells in the spleen (Fig. 1l, m), suggesting KLRG1[-] effector CD8[+] T cells may preferentially traffic into the skin during an acute viral infection. KLRG1 expression was not altered by cognate antigen recognition and as a result, the majority of IFNγ-YFP expressing P14 CD8[+] T cells did not express KLRG1(Fig. 1n, o). Expression of Ki67 by antigen-specific CD8[+] T cells was similar within both VacV-GP33 and VacV-infected skin (Supplementary Fig. 2b, c), consistent with our previous observation that the presence of antigen within the skin microenvironment does not cause significant secondary proliferation of effector CD8[+] T cells[24]. Finally, the presence of cognate antigen within the VacV-infected skin microenvironment significantly enhanced the subsequent formation of CD69[+]CD103[+] $T_{RM}$ CD8[+] T cells (Fig. 1p–r), demonstrating that local antigen is required for IFNγ expression by effector CD8[+] T cells during infection, as well as the subsequent formation of $T_{RM}$ CD8[+] T cells following viral clearance.

Because only a subset of CD8[+] T cells expressed IFNγ in the skin on day 7 after infection, we next tested whether all of the effector CD8[+] T cells isolated from the skin had the potential to express IFNγ in

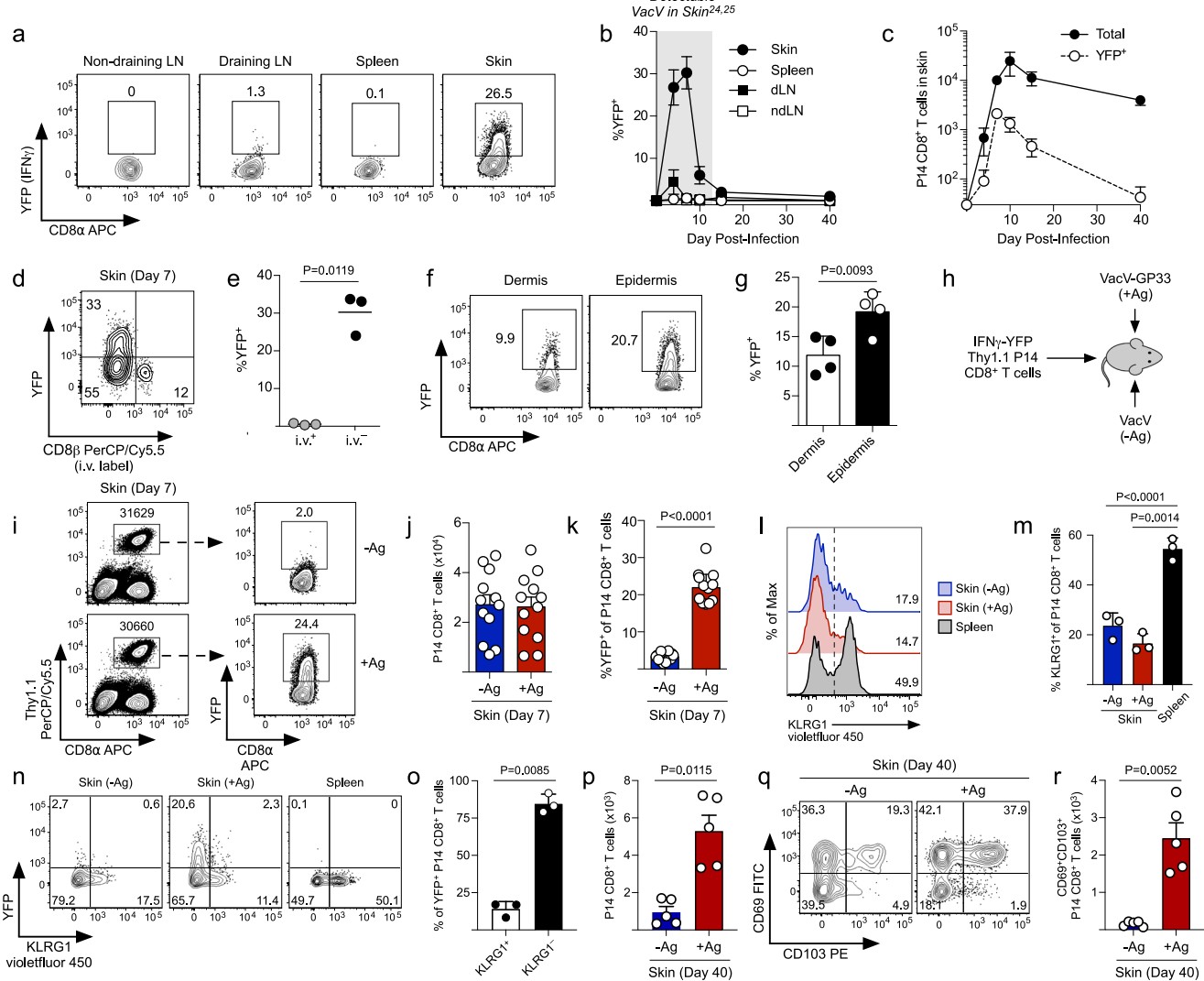

**Fig. 1 | Antigen-specific CD8⁺ T cells transiently express IFNγ in the skin during viral infection in an Ag-dependent manner. a** Naïve IFNγ-YFP P14 CD8⁺ T cells were transferred into B6 mice and infected on the ear skin with VacV-GP33. YFP expression in the indicated organs was quantified by flow cytometry on day 7 post infection. **b** Quantification of (**a**) at the indicated timepoints after infection; $n = 3$ per tissue per time point. **c** Quantification of the total number and YFP⁺ P14 CD8⁺ T cells over time; $n = 3$. **d** Same as (**a**) except CD8β antibody was injected intravenously before sacrifice to distinguish YFP-P14 CD8⁺ T cells within the skin from those in the vasculature. **e** Quantification of (**d**); $n = 3$. **f** Same as (**a**) except YFP expression was quantified by P14 CD8⁺ T cells in the dermis and epidermis on day 7 post infection. **g** Quantification of (**f**); $n = 4$. **h** Experiment design for (**i–r**). **i** Representative flow plots depicting the total number of P14 CD8⁺ T cells and YFP expression by IFNγ-YFP P14 CD8⁺ T in the skin on day 7 post infection.

**j** Quantification of the total number of P14 CD8⁺ T cells in (**i**); $n = 12$. **k** Quantification of YFP expression in (**i**); $n = 12$. **l** Representative histograms depicting the expression of KLRG1. **m** Quantification of (**l**); $n = 3$. **n** Representative flow plots depicting the expression of YFP and KLRG1. **o** Quantification of (**n**); $n = 3$. **p** Quantification of the total number of IFNγ-YFP P14 CD8⁺ T cells in the skin on day 40 post infection; $n = 5$. **q** Representative flow plots depicting the expression of CD69 and CD103 on day 40 post infection. **r** Quantification of (**q**); $n = 5$. Data shown are mean ± SD and representative of 2 or more independent experiments except (j,k) which are cumulative data from 3 independent experiments. Statistical significance was calculated using a paired two-sided t-test (**e, g, k, o, p, r**) or one-way ANOVA followed by Tukey's multiple comparisons test (**m**). Source data are provided as a Source Data file.

response to TCR stimulation. To do this, we cultured effector P14 CD8⁺ T cells from the spleen, VacV-infected skin, and VacV-GP33 infected skin with or without saturating concentrations of GP₃₃₋₄₁. Interestingly, a large portion (~70%) of P14 CD8⁺ T cells in VacV-GP33-infected skin cultured as a single cell suspension began to express IFNγ in the absence of any additional peptide, suggesting that spatiotemporal dynamics within the VacV-GP33-infected skin microenvironment likely limits the ability of CD8⁺ T cells to interact with antigen-presenting cells. However, essentially all P14 CD8⁺ T cells from both VacV-infected skin or VacV-GP33-infected skin, as well as from the spleen, became YFP⁺ after stimulation with GP₃₃₋₄₁ peptide (Supplementary Fig. 2d, e), demonstrating that antigen-specific 'bystander' CD8⁺ T cells have the

full potential to express IFNγ, but are likely not actively engaging cognate pMHC-I within VacV-infected skin. Taken together, these results demonstrate that IFNγ expression identifies the fraction of effector CD8⁺ T cells that are actively engaging antigenic peptide within the VacV-infected skin microenvironment.

### TCR-signaling strength within the skin microenvironment regulates expression of IFNγ and subsequent T_RM CD8⁺ T cell formation

The previous data demonstrated that the presence of cognate antigen was required for both IFNγ expression and optimal T_RM CD8⁺ T cell formation. However, this system tested the complete presence or

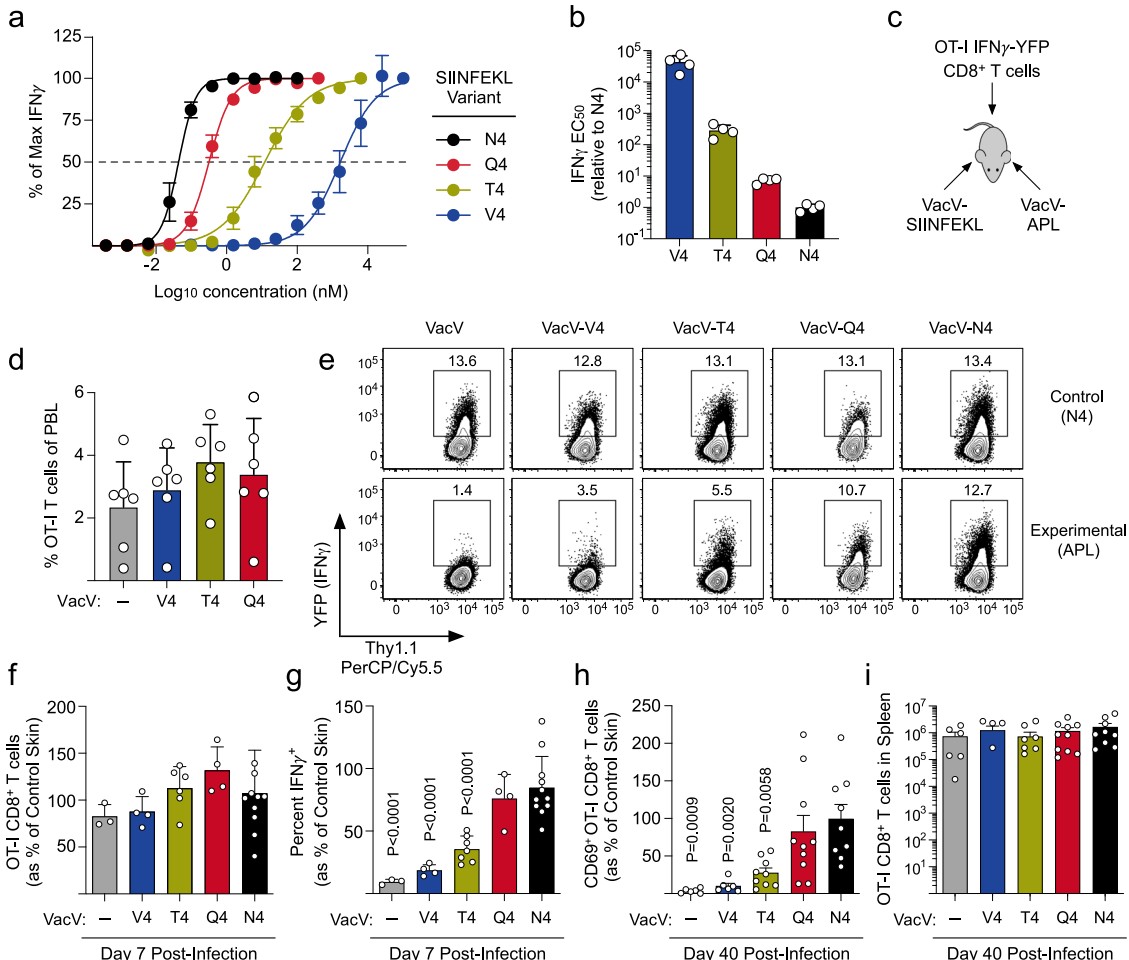

**Fig. 2 | TCR signal strength dictates the magnitude of IFNγ expression and the extent of T_RM CD8+ T cell differentiation. a** Naïve IFNγ-YFP OT-I CD8+ T cells were transferred into B6 mice and infected on the left ear skin with VACV-SIINFEKL. On day 7 post infection, total splenocytes containing IFNγ-YFP OT-I CD8+ T cells were stimulated with APL variants at the indicated concentrations and IFNγ-YFP expression was quantified by flow cytometry. Data are normalized to frequency of IFNγ+ OT-I T cells at the highest peptide concentration. **b** Quantification of the IFNγ half maximal effective concentration (EC50) for the indicated APL; $n = 4$. **c** Experimental design for (**d**–**i**). **d** Quantification of OT-I CD8+ T cells in the blood on day 7 post co-infection; $n = 6$. **e** Representative flow plots depicting IFNγ-YFP expression in the control and experimental skin on day 7 post infection.

**(f)** Quantification of the number of OT-I CD8+ T cells in the skin from (**e**); $n = 3(-)$, 4(V4), 6(T4), 4(Q4), and 11(N4). **g** Quantification of IFNγ-YFP expression by OT-I CD8+ T cells from (**e**); $n = 3(-)$, 4(V4), 6(T4), 4(Q4), and 11(N4). **h** CD69+ OT-I CD8+ T cells in the skin were quantified as a percent of control skin on day 40 post infection; $n = 6(-)$, 6(V4), 9(T4), 10(Q4), and 9(N4). **i** Number of OT-I CD8+ T cells in the spleens from (**h**); $n = 6(-)$, 4(V4), 7(T4), 10(Q4), and 9(N4). Data in (**f**–**h**) are normalized to control (N4-infected) skin. Data shown are mean ± SEM and representative of 2 or more independent experiments except (**f**–**i**) which are cumulative data from 3 independent experiments. Statistical significance was calculated using a one-way ANOVA followed by Dunnett's multiple comparisons test using VacV-N4 as the control group (**g**, **h**). Source data are provided as a Source Data file.

absence of cognate antigen and did not address whether there is a threshold of TCR signal strength that is required for IFNγ expression and/or T_RM differentiation. To test this, we utilized the OT-I TCR-tg CD8+ T cell system in conjunction with a series of well-characterized amino acid variants at the 4th position of SIINFEKL that exhibit decreased affinity for the OT-I TCR[30]. To test the sensitivity of the OT-I TCR to these altered peptide ligands (APLs), we isolated effector OT-I CD8+ T cells from the spleen on day 7 post VacV-SIINFEKL infection and quantified the concentration of peptide required for a half-maximal IFNγ-YFP response (EC50). Similar to what has been reported previously[30], IFNγ-YFP OT-I CD8+ T cells exhibited a wide range of sensitivities to the APLs (Fig. 2a, b), demonstrating that the same concentration of peptide leads to different levels of OT-I TCR engagement and subsequent IFNγ expression.

We reasoned that infection with VacV strains expressing these SIINFEKL APLs could be used to vary the strength of TCR stimulation for OT-I CD8+ T cells without changing the overall inflammatory milieu or antigen load within the VacV-infected skin

microenvironment. To this end, we generated VacV that expressed SIINFEKL or one of the lower affinity variants. Because the strength of TCR stimulation directly impacts the degree of CD8+ T cell proliferation, we first tested the functionality of VacV-APLs by measuring activation of naive OT-I CD8+ T cells within the draining lymph node. Each VacV strain was equally infectious (Supplementary Fig. 3a), but cellular proliferation and expression of the activation markers CD25 and CD69 (expressed before cell division begins) correlated with APL affinity (Supplementary Fig. 3b–e), demonstrating that the low-affinity variant peptides are expressed and presented. This decreased level of activation resulted in lower frequencies of effector OT-I CD8+ T cells in the circulation (Supplementary Fig. 3f), confirming that VacV-APLs differentially activate naïve OT-I CD8+ T cells.

To control for different levels of activation by the VacV-APLs, we used a co-infection system where the left ear skin was infected with VacV-SIINFEKL (control skin) and the right ear skin is infected with one of the VacV-APL strains (experimental skin; Fig. 2c). In contrast to

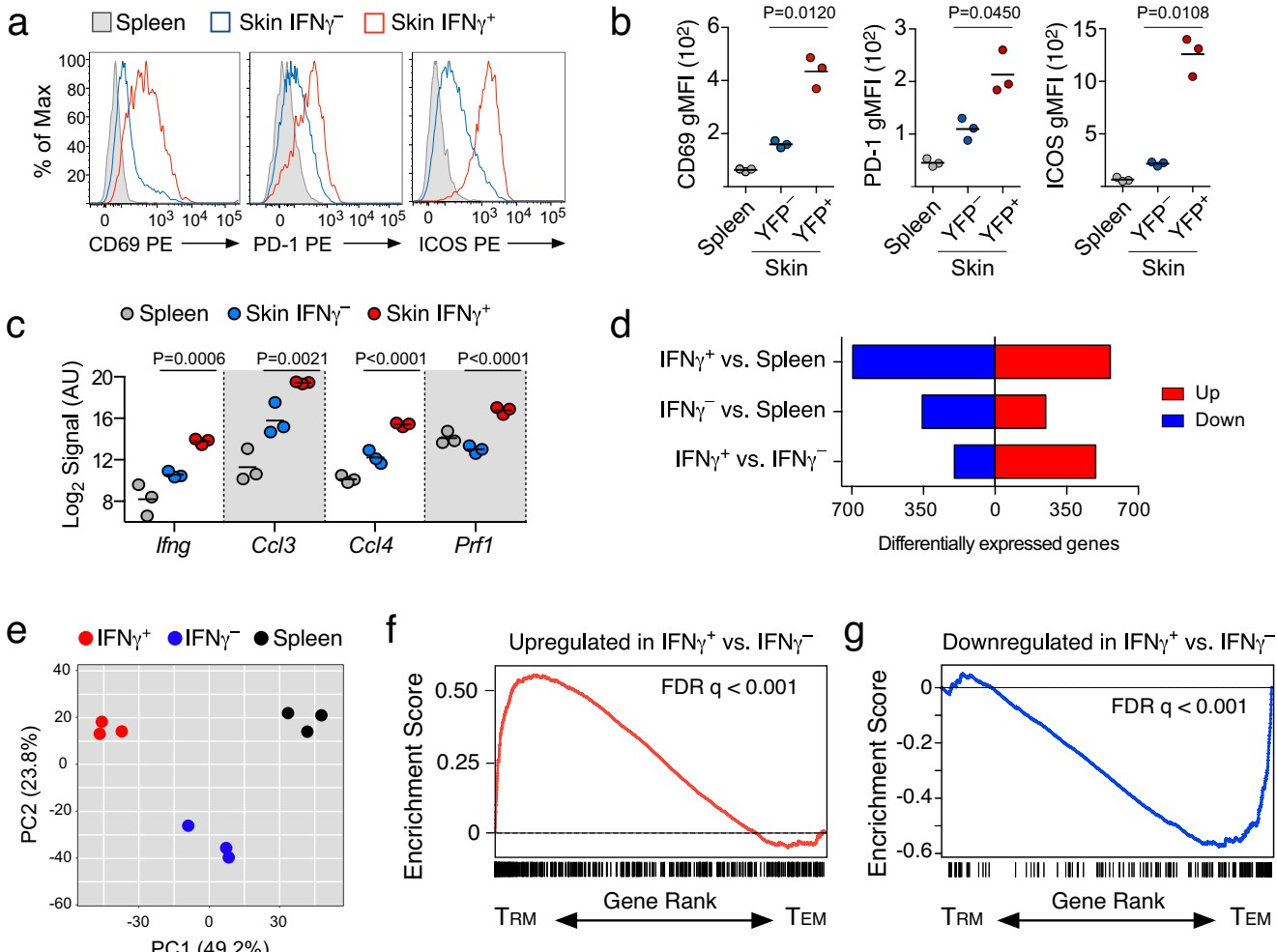

**Fig. 3 | CD8$^+$ T cells producing IFNγ within the skin microenvironment exhibit phenotypic and transcriptional features of T$_{RM}$ CD8$^+$ T cells.** Naïve IFNγ-YFP P14 CD8$^+$ T cells were transferred into B6 mice and infected on the left ear skin with VacV-GP33. On day 7 post infection, effector P14 CD8$^+$ T cells were sorted from the skin and spleen based on YFP expression. **a** Representative histograms depicting the expression of phenotypic features of IFNγ$^+$, IFNγ$^-$, or spleen effector P14 CD8$^+$ T cells on day 7 post infection. **b** Quantification of the (**a**); $n = 3$ per group. **c**–**g** The transcriptional profile of sorted T cells was analyzed by microarray. **c** The expression levels of the indicated genes from microarray analysis; $n = 3$. **d** The number of differentially expressed genes between the sorted subsets. **e** Principal component analysis of the sorted subsets. **f, g** Gene Set Enrichment Analysis (GSEA) comparing genes up- or down-regulated in IFNγ$^+$ cells to transcriptional profiles of mature VacV-specific T$_{RM}$ or T$_{EM}$ CD8$^+$ T cell populations[33]. Data from (**a, b**) are representative of 2 independent experiments and statistical significance was performed using a two-sided paired t-test. Statistical significance (**c**) was performed using a one-way ANOVA followed by Tukey's multiple comparisons test. Source data are provided as a Source Data file.

single infections, the magnitude of the effector T cell response was equivalent in co-infected mice (Fig. 2d), suggesting that the VacV-SIINFEKL infection was the dominant driver of OT-I CD8$^+$ T cell activation and expansion using this model. OT-I CD8$^+$ T cells were recruited equally to both the control (N4-infected) and experimental (APL-infected) skin on day 7 post infection, but the frequency of OT-I CD8$^+$ T cells expressing IFNγ directly correlated with the strength of TCR signal received within the VacV-infected skin microenvironment (Fig. 2e–g). On day 40 post infection, the number of OT-I T$_{RM}$ CD8$^+$ T cells was reduced in skin where lower affinity peptides were presented (Fig. 2h), suggesting that the strength of TCR stimulation required for IFNγ expression in the skin is largely equivalent to that required for subsequent T$_{RM}$ CD8$^+$ T cell differentiation, whereas the number of memory OT-I CD8$^+$ T cells in the circulation was not impacted (Fig. 2i). Together, these data demonstrate that the strength of TCR stimulation received within the VacV-infected skin microenvironment is a critical regulator for both the execution of effector functions (e.g., production of cytokines) and the subsequent development of T$_{RM}$ CD8$^+$ T cells.

### CD8$^+$ T cells producing IFNγ in the skin exhibit phenotypic and transcriptional features of T$_{RM}$ differentiation

Multiple studies have identified gene sets that are expressed by developing and/or mature skin-T$_{RM}$ CD8$^+$ T cells[14,31–33]. Following VacV infection, nearly all IFNγ$^+$ CD8$^+$ T cells in the skin expressed high levels of CD69, PD-1, and ICOS, all of which are phenotypic features of T$_{RM}$ CD8$^+$ T cells (Fig. 3a, b). To further test whether T cells receiving TCR stimulation within VacV-infected skin were undergoing T$_{RM}$ differentiation, we sort purified TCR-stimulated effector CD8$^+$ T cells (IFNγ$^+$) cells from the skin, antigen-specific bystander T cells from the same skin microenvironment (IFNγ$^-$), and effector T cells that have not been exposed to the skin microenvironment or received a second antigen encounter (spleen) on day 7 postinfection (Supplementary Fig. 4a) and generated genome-wide transcriptional profiles. IFNγ$^+$ T cells had higher transcript levels of *Ifng* as well as other TCR-dependent effector genes including *Ccl3*, *Ccl4*, and *Prf1* (Fig. 3c), offering further evidence that IFNγ-YFP$^+$ T cells in the skin are exhibiting multiple TCR-dependent effector functions. Several hundred genes were differentially expressed among the three T cell populations and the transcriptional profiles

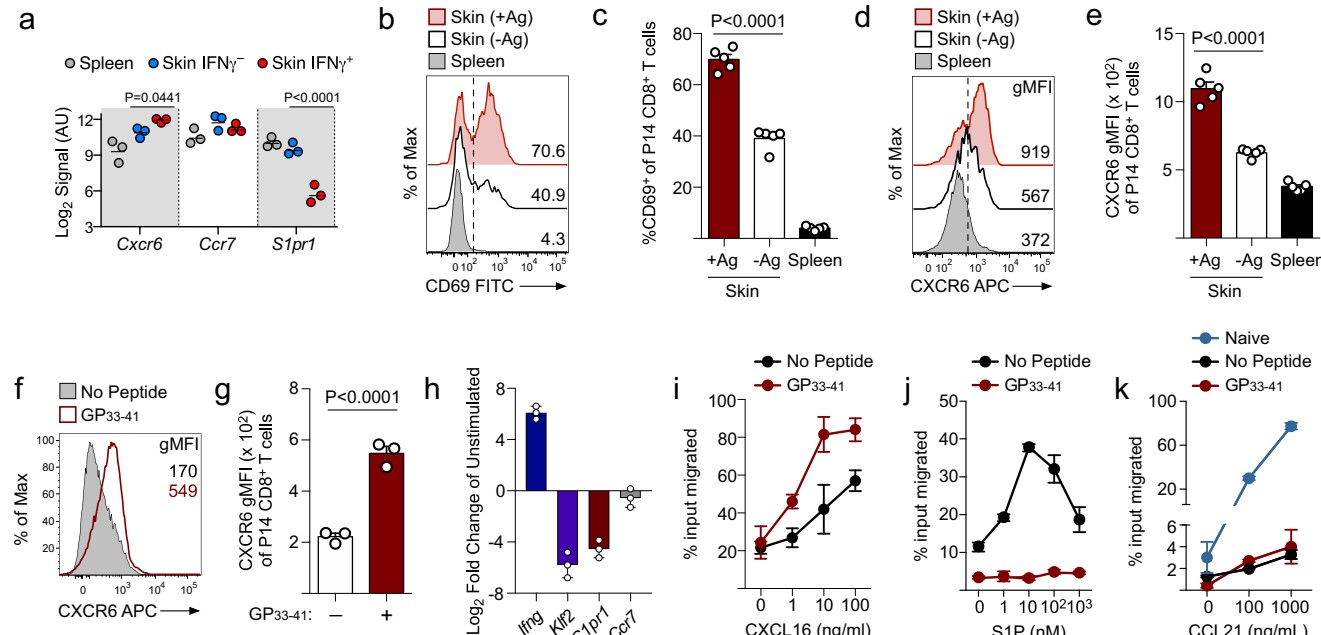

**Fig. 4 | TCR stimulation changes the chemotactic properties of effector CD8+ T cells. a** Expression of the indicated chemokine receptors from the microarray analysis performed in Fig. 3; $n = 3$. **b**–**e** Naïve P14 CD8+ T cells were transferred into B6 mice and infected with VacV-GP33 (+Ag) and VacV (-Ag) on the left and right ear skin, respectively. **b** Representative histograms depicting expression of CD69 by effector P14 CD8+ T cells within the skin and spleen on day 15 post infection. **c** Quantification of (**b**); $n = 5$. **d** Representative histograms depicting the expression CXCR6 by effector P14 CD8+ T cells within the skin and spleen on day 15 post infection. **e** Quantification of (**d**); $n = 5$. **f**–**k** Mice received naïve P14 CD8+ T cells followed by infection with VacV-GP33 on the left ear skin. On day 10 post-infection, total splenocytes were stimulated with 10 nM GP$_{33-41}$ ex vivo for 18 or 48 hours at 37 °C. **f** Representative histograms depicting CXCR6 expression following stimulation of effector P14 CD8+ T cells with GP$_{33-41}$. **g** Quantification of (f); $n = 3$. **h** The

expression level of the indicated genes was quantified by qPCR; $n = 3$ **i** Migration of effector P14 CD CD8+ T cells from the spleen in response to the indicated concentrations of CXCL16 following stimulation with GP$_{33-41}$; $n = 3$. **j** Migration of effector P14 CD8+ T cells from the spleen in response to the indicated concentrations of S1P following stimulation with GP$_{33-41}$; $n = 2$ per dilution per group. **k** Migration of effector P14 CD8+ T cells from the spleen in response to the indicated concentrations of CCL21 following stimulation with GP$_{33-41}$; $n = 2$ per dilution per group. Naïve CD8+ T cells were included as positive migration control for CCL21. Migration of T cells was calculated as percentage migrated of the total input of P14 CD8+ T cells. Data shown are mean ± SD. **b**–**k** are representative of 2 or more independent experiments. Statistical significance was calculated using a one-way ANOVA followed by Tukey's multiple comparisons test (a, c, e) or a two-sided paired t-test (g). Source data are provided as a Source Data file.

were clearly distinct based on both principal component analysis and hierarchical clustering (Fig. 3d, e and Supplementary Fig. 4b, c). These data also show that there are significant changes in gene expression associated with both entry into the tissue microenvironment (Spleen vs. IFNγ⁻), as well as subsequent TCR stimulation within the skin.

In order to test in a quantitative fashion whether TCR stimulation within the context of the VacV-infected skin microenvironment promoted T$_{RM}$ differentiation, we performed gene set enrichment analysis (GSEA) comparing the set of genes that were up- or down-regulated in IFNγ⁺ T cells to the published expression profile of mature VacV-specific T$_{RM}$ and circulating effector memory (T$_{EM}$) CD8+ T cell populations[33]. Strikingly, GSEA showed that the set of genes upregulated in IFNγ⁺ T cells were also more highly expressed in mature T$_{RM}$ populations (Fig. 3f), and the set of genes downregulated in IFNγ⁺ cells were more highly expressed in circulating T$_{EM}$ populations (Fig. 3g). These data demonstrate that effector T cells receiving TCR stimulation within the skin microenvironment have already undergone changes in gene expression that resemble mature T$_{RM}$ CD8+ T cells, whereas previous studies have reported that the T$_{RM}$ transcriptional program is not fully engaged until >25 days postinfection when analyzing bulk T cell populations within the skin[33]. Thus, TCR engagement within the skin promoted expression of genes necessary for T$_{RM}$ differentiation and retention, while repressing gene networks that promote the development of circulating memory CD8+ T cells. Together, these data demonstrate that IFNγ⁺, IFNγ⁻, and splenic effector CD8+ T cells are three distinct populations, and of those three populations, IFNγ⁺ T cells uniquely exhibit documented features of mature T$_{RM}$ CD8+ T cells.

## Antigen recognition changes the chemotactic properties of effector CD8+ T cells

Once within nonlymphoid tissues, effector CD8+ T cells are subjected to a variety of opposing chemotactic gradients that ultimately dictate tissue-retention or a return to the circulation by being drawn into draining lymphatic vessels[34]. Our gene expression analysis showed that within the skin, IFNγ⁺ T cells exhibited higher expression of *Cxcr6* and lower expression of *S1pr1*, whereas expression of *Ccr7* was comparable (Fig. 4a), suggesting that TCR engagement may partly regulate T$_{RM}$ differentiation by modulating differential expression of specific chemokine receptors on CD8+ T cells within the skin. To test whether antigen recognition was necessary to alter chemokine receptor expression of CD8+ T cells in the skin, naïve TCR-tg P14 CD8+ T cells were transferred into B6 mice followed by co-infection with VacV (+/-Ag). Effector P14 CD8+ T within +Ag skin exhibited greater expression of CD69 and CXCR6 compared with T cells within -Ag skin or spleen (Fig. 4b–e), demonstrating that local antigen recognition is necessary to fully engage the tissue-retention program in effector CD8+ T cells within the VacV-infected skin microenvironment.

To further investigate whether secondary antigen encounter was sufficient to change the chemotactic properties of effector CD8+ T cells, we stimulated effector P14 CD8+ T cells from the spleen with GP$_{33-41}$. Consistent with our gene expression profile, TCR stimulation increased expression of CXCR6 and downregulated expression of *S1pr1* along with its transcriptional regulator *Klf2*[35], whereas expression of *Ccr7* remained unchanged (Fig. 4f–h). To test the functional significance of antigen-dependent changes in the chemokine receptor

profile of effector CD8$^+$ T cells, chemotaxis assays were performed. In accordance with upregulated CXCR6 surface expression, down-regulated *S1pr1* gene expression and unchanged *Ccr7* gene expression, ex vivo TCR stimulation increased migration to CXCL16 (ligand of CXCR6), completely abolished migration to S1P (ligand of S1PR1) and did not change the low level of migration observed in response to CCL21 (ligand of CCR7) when compared with unstimulated effector CD8$^+$ T cells (Fig. 4i–k), demonstrating that TCR stimulation of effector CD8$^+$ T cells is essential to prevent S1P-mediated egress and promote CXCR6-mediated retention in the skin. Taken together, these data demonstrate that antigen encounter by effector CD8$^+$ T cells is fundamental to engage a unique chemotaxis profile that favors retention and migration within the skin microenvironment.

### Strength of TCR signaling regulates Blimp1 expression and the chemotactic properties of effector CD8$^+$ T cells

Blimp1 is a transcription factor encoded by the gene *Prdm1* and has been shown to promote tissue residency (with its homolog *Hobit*) in some lymphocyte populations[9,36], but the mechanisms that regulate Blimp1 expression, particularly within nonlymphoid tissues, remain unclear. In our transcriptional analysis, IFNγ$^+$ T cells displayed higher expression of *Prdm1* (Fig. 5a), suggesting a possible mechanism directly linking TCR stimulation to T$_{RM}$ CD8$^+$ T cell differentiation. Notably, we did not identify *Hobit* as being differentially expressed between IFNγ$^+$ and IFNγ$^-$ T cells from the skin compared with effector T cells from the spleen. We confirmed increased expression of Blimp1 at the protein level by flow cytometry and found higher Blimp1 expression within IFNγ$^+$ T cells (Fig. 5b, c). Thus, these data suggest that expression of Blimp1 by effector CD8$^+$ T cells may be directly linked to secondary antigen recognition in the periphery.

ICOS is a co-stimulatory molecule that has also been reported to be consistently upregulated among T$_{RM}$ signature genes and selective engagement of ICOS on antigen-specific effector CD8$^+$ T cells was found to be critical for T$_{RM}$ differentiation within nonlymphoid tissues[37]. Interestingly, similar to expression of IFNγ, elevated ICOS expression by ~25% of CD8$^+$ T cells was observed in +Ag skin and ICOS$^+$ T cells also expressed higher levels of Blimp1 compared with ICOS$^-$ effector T cells from both the skin and spleen (Supplementary Fig. 5a–c). To measure both the antigen-dependence, as well as the kinetics of Blimp1/ICOS expression during viral skin infection, we quantified the frequency of effector P14 CD8$^+$ T cells co-expressing ICOS and Blimp1 (ICOS$^+$Blimp1$^+$) at different time points following VacV skin infection (+/- Ag). The frequency of ICOS$^+$Blimp1$^+$ P14 CD8$^+$ T cells was significantly higher on CD8$^+$ T cells in +Ag skin on day 7 post infection (Supplementary Fig. 5d, e). However, expression of both ICOS and Blimp1 was rapidly lost coincident with viral clearance suggesting that TCR stimulation is critical to trigger the transcription of Blimp1 and ICOS and that skin T$_{RM}$ programming occurs primarily during this window of antigen-specific encounters. Together, these data identify IFNγ, ICOS, and Blimp1 as functional and phenotypic markers that are concomitantly upregulated by effector CD8$^+$ T cells receiving a second TCR stimulation in the periphery.

Because expression of Blimp1 was found to be enriched only in +Ag skin, we next asked whether secondary antigen encounter would be sufficient to promote Blimp1 expression. Effector P14 CD8$^+$ T cells from the spleen stimulated with GP$_{33-41}$ caused Blimp1 to be expressed, but was completely blocked by the addition of low concentrations of FK506, an immunosuppressant that blocks calcineurin-mediated dephosphorylation and activation of the transcription factor nuclear factors of activated T cells (NFAT)[38] (Fig. 5d, e) suggesting that TCR-mediated expression of Blimp1 is dependent on canonical TCR signaling resulting in NFAT-mediated gene transcription. Interestingly, whereas effector P14 CD8$^+$ T cells stimulated with increasing concentrations of GP$_{33-41}$ peptide was sufficient to upregulate Blimp1 expression in a dose dependent manner (Fig. 5f, g), naïve CD8$^+$ T cells

did not express Blimp1 even at the highest concentration of peptide tested, suggesting that elevated Blimp1 expression is uniquely engaged following a 'second' antigen encounter and not during the initial activation of naïve CD8$^+$ T cells. In contrast, both naïve and effector CD8$^+$ T cells readily expressed CD69 when stimulated with peptide (Fig. 5h, i). Together, these data suggest that the initial priming of CD8$^+$ T cells is not sufficient to trigger heightened Blimp1 expression and a second antigen encounter is required to fully upregulate the Blimp1 transcription factor within effector CD8$^+$ T cells in the periphery.

Our findings demonstrated that strength of TCR signaling was responsible for the extent of T$_{RM}$ differentiation during viral skin infection, thus, we next tested whether Blimp1 expression was also dependent on high-affinity peptide recognition. We stimulated effector OT-I CD8$^+$ T cells with increasing concentrations SIINFEKL APLs, and found that similar to IFNγ expression (Fig. 2a, b), expression of Blimp1, CD69, and ICOS were all also proportionate to TCR affinity and signal strength (Fig. 5k–m and Supplementary Fig. 5f, g). Furthermore, stimulation with native SIINFEKL completely abolished migration of effector OT-I CD8$^+$ T cells to S1P, while OT-I CD8$^+$ T cells stimulated with the lower affinity SIINFEKL variants exhibited S1P-mediated migration that negatively correlated with Blimp1 and CD69 expression (Fig. 5n), demonstrating that strong TCR signaling is required to fully suppress the S1P-dependent egress pathway. Interestingly, the expression level of CXCR6 was similarly induced by SIINFEKL variants regardless of signal strength, which was confirmed by largely equal migration of effector OT-I CD8$^+$ T cells stimulated by various APLs in response to CXCL16 (Fig. 5o–q) suggesting that low affinity agonists are sufficient to promote CXCR6-mediated migration, but that high-affinity agonists are necessary to prevent S1P-mediated migration. Together, these data demonstrate that antigen recognition within the skin microenvironment is a primary determinant of Blimp1 and CXCR6 expression and directly contributes to suppression of the S1P-mediated egress pathway.

### Tissue-resident memory CD8$^+$ T cell differentiation in the skin requires both antigen recognition and Blimp1 expression

After observing that Blimp1 was strongly upregulated when effector CD8$^+$ T cells re-encountered their cognate antigen, we next tested whether Blimp1 would be necessary for CD8$^+$ T cells to become T$_{RM}$ during a viral skin infection. WT (*Prdm1$^{+/+}$*ROSA26-Cre-ER$^{T2}$; Thy1.1/1.1) and Blimp1$^{-/-}$ (*Prdm1$^{fl/fl}$* ROSA26-Cre-ER$^{T2}$; Thy1.1/1.2) naïve P14 CD8$^+$ T cells were co-transferred into B6 mice and tamoxifen administration began two days before infection with VacV (+/- Ag) on the left and right ear skin (Fig. 6a). The loss of Blimp1 protein expression following tamoxifen treatment was confirmed by flow cytometry following ex vivo stimulation with GP$_{33-41}$ peptide (Supplementary Fig. 6a, b). Through day 15 postinfection, accumulation of WT and Blimp1$^{-/-}$ CD8$^+$ T cells in the skin was largely similar and was not influenced by local antigen recognition (Supplementary Fig. 7). In the circulation, Blimp1$^{-/-}$ CD8$^+$ T cells exhibited a strong T$_{CM}$ phenotype as evidenced by higher expression of both CD62L and TCF-1 compared with WT CD8$^+$ T cells (Fig. 6b–e), as predicted[39]. In agreement with heightened CD62L expression, Blimp1$^{-/-}$ CD8$^+$ T cells were found at higher frequencies than WT T cells in lymph nodes (Fig. 6f), but were present at similar frequencies in the skin and in the circulation.

Previous studies reported that Ag-recognition in the skin caused CD8$^+$ T cells to express CD69 and become tissue-resident[24], thus, we next asked whether Blimp1 was functioning downstream of TCR signaling to regulate CD69 expression in the skin. Antigen-recognition in the skin caused WT CD8$^+$ T cells to express more CD69 compared with skin where cognate antigen was not present. Blimp1$^{-/-}$ CD8$^+$ T cells failed to upregulate expression of CD69 in an Ag-dependent manner and was largely similar to CD69 expression found in skin infected with VacV not expressing cognate antigen (Fig. 6g, h). In contrast,

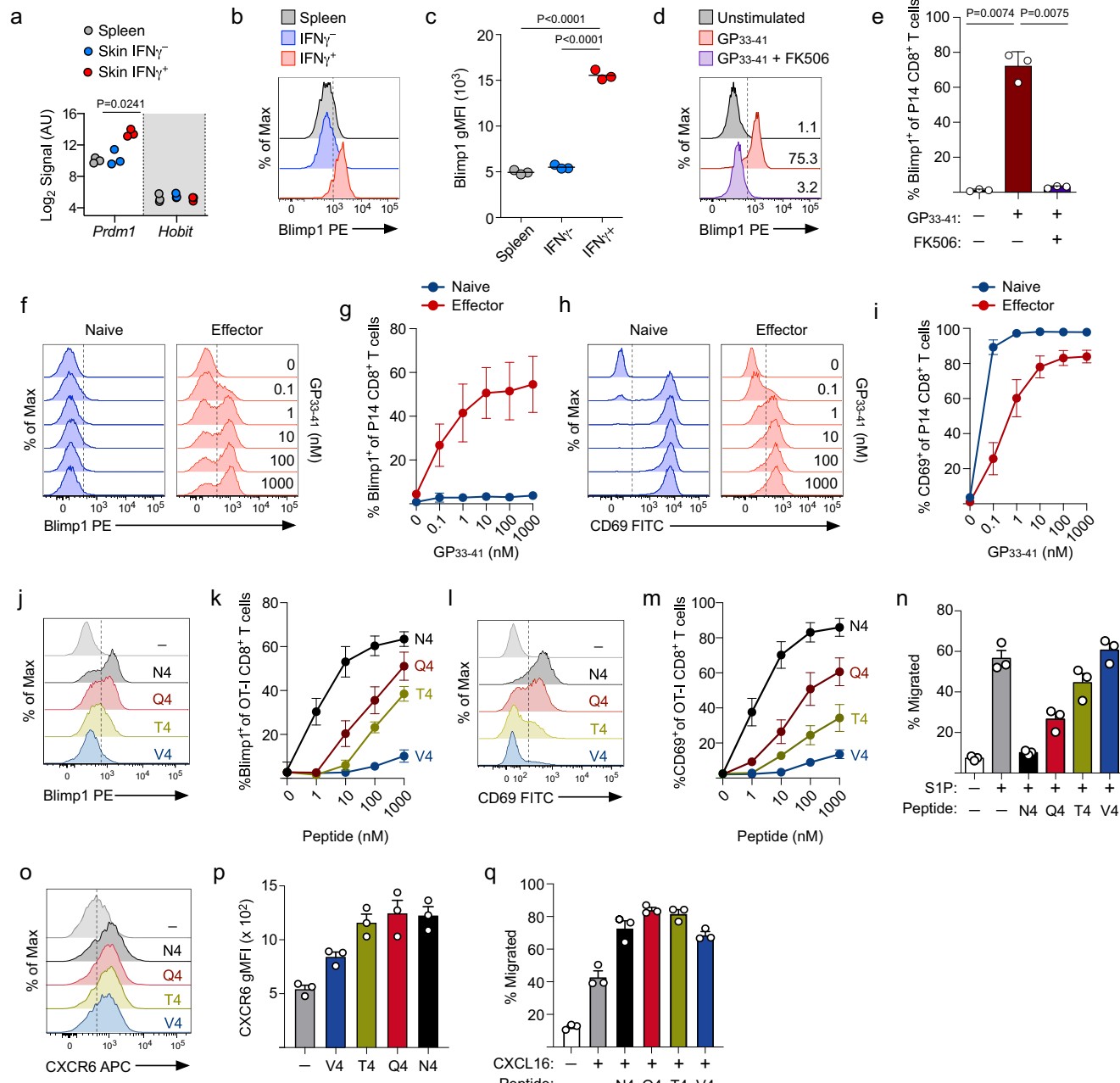

**Fig. 5 | Strength of TCR signaling regulates Blimp1 expression and the chemotactic features of effector CD8⁺ T cells. a** *Prdm1* and *Hobit* expression from the microarray analysis performed in Fig. 3; *n* = 3. **b** Naïve IFNγ-YFP P14 CD8⁺ T cells were transferred into B6 mice and infected on the left ear skin with VacV-GP33. Representative histograms depicting the protein expression of Blimp1 on the indicated subsets of effector P14 CD8⁺ T cells. **c** Quantification of (**b**); *n* = 3. **d**–**i** Naïve P14 CD8⁺ T cells were transferred into B6 mice and infected on the left ear skin with VacV-GP33. On day 10 post-infection, splenocytes containing P14 CD8⁺ T cells were stimulated ex vivo with GP33-41. **d** Representative histogram depicting expression of Blimp1 following stimulation with 10 nM GP33-41 with or without FK506. **e** Quantification of (**d**); *n* = 3. **f** Representative histograms depicting Blimp1 expression following stimulation of splenocytes containing either naïve or effector P14 CD8⁺ T cells with increasing concentration of GP33-41. **g** Quantification of (**f**); *n* = 5. **h** same as in (**f**), but expression of CD69 is displayed. **i** Quantification of (**h**);

*n* = 5. **j**–**q** Naïve OT-I CD8⁺ T cells were transferred into B6 mice and infected on the left ear skin with VacV-SIINFEKL. On day 10 post infection, splenocytes containing effector OT-I CD8⁺ T cells were stimulated ex vivo. **j** Representative histograms depicting expression of Blimp1 by effector OT-1 CD8⁺ T cells following stimulation with 10 nM APLs for 18 hours. **k** Quantification of (**j**); *n* = 5. **l** Same as in (**j**), but the expression of CD69 is displayed. **m** Quantification of (**l**); *n* = 5. **n** Migration of effector OT-I CD8⁺ T cells in response to 50 nM S1P following stimulation with 10 nM APLs; *n* = 3. **o** Same as in (**j**), but expression of CXCR6 is displayed. **p** Quantification of (**o**); *n* = 3. **q** Same as in (**n**), but migration was quantified in response to 10 ng/ml CXCL16; *n* = 3. Data shown are mean ± SD and representative of 2 or more independent experiments. Statistical significance was calculated using a one-way ANOVA followed by Tukey's multiple comparisons test (**e**). Source data are provided as a Source Data file.

expression of CD103, which is thought to require signaling downstream of the TGF-β receptor[18], did not require antigen recognition or Blimp1 expression (Fig. 6i, j). In agreement with this observation, both WT and Blimp1⁻/⁻ effector CD8⁺ T cells expressed CD103 in response to

TGF-β in vitro (Supplementary Fig. 8a, b), demonstrating that distinct signaling pathways control unique features of T_RM differentiation. CD69 expression by Blimp1⁻/⁻ CD8⁺ T cells was lower in both the CD103⁺ and CD103⁻ populations in the skin on day 15 postinfection,

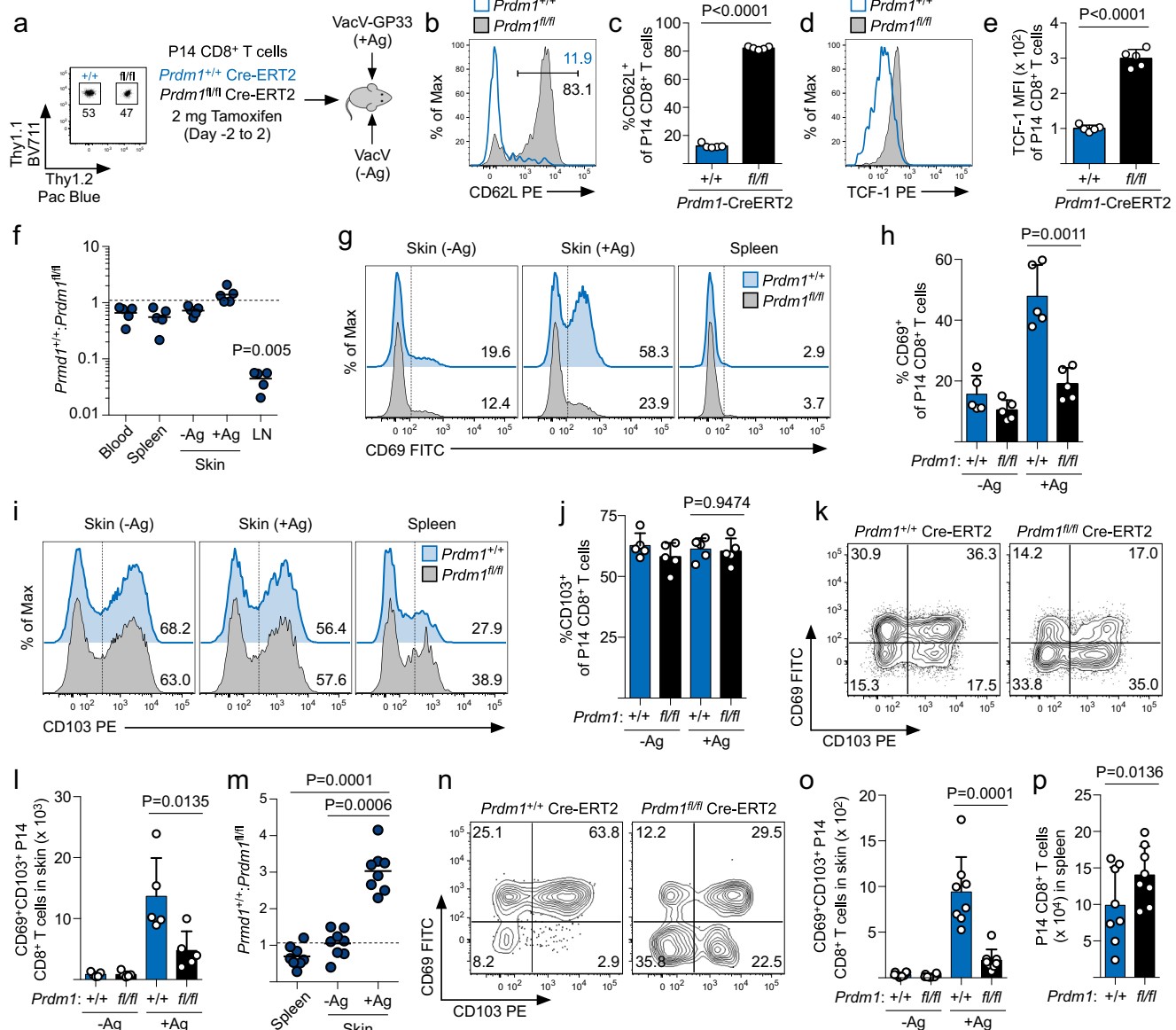

**Fig. 6 | Antigen recognition and Blimp1 expression are both required for CD8⁺ T cells to differentiate into T_RM in the skin following viral infection.**
**a** Experimental design. A - 1:1 mixture of naïve Thy1.1/Thy1.1 *Prdm1*⁺/⁺ ROSA26-Cre-ER^T2 and Thy1.1/1.2 *Prdm1*^fl/fl ROSA26-Cre-ER^T2 P14 CD8⁺ T cells were transferred into B6 mice and were co-infected with VacV-GP33 and VacV on the left and right ear skin, respectively. Tamoxifen treatment was initiated one day prior to the T cell transfer and continued for 5 consecutive days. **b** Representative histograms depicting the expression of CD62L by effector P14 CD8⁺ T cells on day 15 post infection. **c** Quantification of (**b**); *n* = 5. **d** Same as in (**b**), but expression of the TCF-1 is displayed. **e** Quantification of (**d**); *n* = 5. **f** Ratio of *Prdm1*⁺/⁺ and *Prdm1*^fl/fl effector P14 CD8⁺ T cells on day 15 post-infection; *n* = 5. **g** Representative histograms depicting the expression of CD69 by P14 CD8⁺ T cells in the spleen and skin on day 15 post infection. **h** Quantification of (**g**); *n* = 5. **i** Same as in (**g**), but the expression of CD103 is displayed. **j** Quantification of (**i**); *n* = 5. **k** Representative flow plots depicting the expression of CD69 and CD103 by P14 CD8⁺ T cells on day 15 post infection. **l** Quantification of the number of CD69⁺CD103⁺ P14 CD8 T cells in (**k**); *n* = 5. **m** Ratio of *Prdm1*⁺/⁺ and *Prdm1*^fl/fl memory P14 CD8⁺ T cells on day 40 post-infection; *n* = 8. **n** Representative flow plots depicting the expression of CD69 and CD103 by *Prdm1*⁺/⁺ and *Prdm1*^fl/fl P14 CD8⁺ T cells in VacV-GP33 (+Ag) infected skin on day 40 post-infection. **o** Quantification of the number of CD69/CD103⁺ P14 CD8⁺ T cells from (**n**); *n* = 8. **p** Quantification of P14 CD8⁺ T cells in the spleen; *n* = 8. Data shown are mean ± SD and are representative of 2 independent experiments. Statistical significance was calculated using a one-way ANOVA followed by Tukey's multiple comparisons test (**f**, **h**, **j**, **l**, **m**, **o**, **p**) or two-sided paired t-tests (**c**, **e**). Source data are provided as a Source Data file.

resulting in a significant reduction in the number of CD69/CD103⁺ T_RM precursors (Fig. 6k, l). Thus, these data demonstrate that TCR-stimulated expression of Blimp1 is required for effector CD8⁺ T cells to express CD69, but not CD103, in the skin during viral infection.

Because Blimp1⁻/⁻ CD8⁺ T cells were unable to express CD69 following antigen recognition in the skin suggested these T cells may not be retained and fully differentiate into a T_RM population. Indeed, at day 40 postinfection, there were more WT CD8⁺ T cells in +Ag skin than Blimp1⁻/⁻ T cells (Fig. 6m and Supplementary Fig. 7). Blimp1⁻/⁻ CD8⁺

T cells also failed to mature into a CD69⁺CD103⁺ population, resulting in significantly reduced overall number of Blimp1⁻/⁻ T_RM CD8⁺ T cells in the skin (Fig. 6n, o), whereas there were statistically more Blimp1⁻/⁻ CD8⁺ T cells in the spleen compared with WT controls (Fig. 6p). Notably, the small number of CD8⁺ T cells that remain in -Ag skin following the resolution of viral infection did not require Blimp1 (Fig. 6o and Supplementary Fig. 7). Overall, these data show that TCR-stimulated expression of Blimp1 during viral skin infection is necessary for effector CD8⁺ T cells to fully differentiate into a CD69⁺CD103⁺ T_RM population.

**TCR-stimulated expression of Blimp1 is required to establish the chemotactic properties of effector CD8⁺ T cells that support tissue-residency**

Having established that expression of Blimp1 and the chemotactic properties of effector CD8⁺ T cells were both regulated by strength of TCR signaling, we next asked if acquisition of the 'chemotactic switch' was therefore dependent on Blimp1. Expression of both CD69 and CXCR6 (Fig. 7a–d) gradually increased on antigen-specific WT CD8⁺ T cells in VacV-infected skin between days 7 and 15 postinfection, whereas expression of both genes remained low on Blimp1⁻/⁻ T cells, suggesting that TCR stimulated expression of Blimp1 subsequently programs the effector CD8⁺ T cells to become resident by both promoting CXCR6-mediated retention and limiting S1P-mediated egress. WT CD8⁺ T cells from the skin exhibited greater migration towards CXCL16 than Blimp1⁻/⁻ T cells (Fig. 7e) demonstrating that Blimp1 expression largely underlies CXCR6-dependent chemotaxis of effector CD8⁺ T cells within VacV-infected skin. As shown previously, WT effector P14 CD8⁺ T cells from the spleen stimulated with GP₃₃-₄₁ peptide upregulated expression of CXCR6 resulting in enhanced migration towards CXCL16, whereas Blimp1⁻/⁻ T cells remained largely unresponsive to this chemokine even following peptide stimulation (Fig. 7f–h). WT and Blimp1⁻/⁻ effector CD8⁺ T cells from the spleen both exhibited robust and equal migration toward S1P. However, in contrast to the complete lack of S1P-mediated migration observed when WT CD8⁺ T cells were stimulated with peptide, Blimp1⁻/⁻ CD8⁺ T cells still migrated toward S1P following peptide stimulation (Fig. 7i), demonstrating that TCR stimulated expression of Blimp1 is required to both promote CXCR6-mediated migration, while simultaneously suppressing S1P-mediated tissue egress following antigen encounter in the periphery. Overall, these data show that S1P-mediated tissue egress and limited responsiveness to CXCL16 is the default migratory feature of activated effector CD8⁺ T cells after exiting the draining lymph node and that the strength of TCR stimulation necessary for Blimp1

expression is essential to establish the chemotactic properties that support T$_{RM}$ differentiation.

## Discussion

T$_{RM}$ CD8⁺ T cells have been identified and described in a vast array of tissue microenvironments, yet the mechanisms that control their unique forms of differentiation remain unresolved. While multiple studies argue for antigen-independent pathways of T$_{RM}$ differentiation, this work and our previously published studies have clearly demonstrated that the presence of local antigen results in far more robust T$_{RM}$ development than exposure to the tissue microenvironment alone[7,24,25]. By using the IFNγ-YFP approach described here, we were able to compare three populations of effector CD8⁺ T cells; those within the spleen that have not yet been exposed to the tissue environment, T cells in the skin that have not engaged antigenic peptide, and T cells in the skin that are actively receiving TCR stimulation. Thus, this allowed us to directly examine the transcriptional consequences of TCR stimulation compared with exposure to the tissue microenvironment alone, which has been argued to be sufficient to promote T$_{RM}$ differentiation. Consistent with this notion, we found that expression of particular T$_{RM}$-associated genes appears to be regulated by exposure to the VacV-infected skin microenvironment (*Itgae*) and others that are regulated in response to TCR stimulation within VacV-infected skin (*Cxcr6*, *Prdm1*, *S1pr1*, *Icos*). Thus, these data illuminate which pathways are engaged in an antigen-dependent manner and those that are engaged in response to exposure to the inflammatory microenvironment.

Here, we demonstrate that TCR signaling by effector CD8⁺ T cells is a major driver of global changes in gene expression that promote T$_{RM}$ development, including the key target genes *Cxcr6*, *Prdm1*, and *Icos*. Blimp1 is a transcriptional repressor known to recruit epigenetic-silencing factors to the *Il2ra* and *Cd27* loci in effector T cells[40], which are critical steps in the development of long-lived, circulating memory

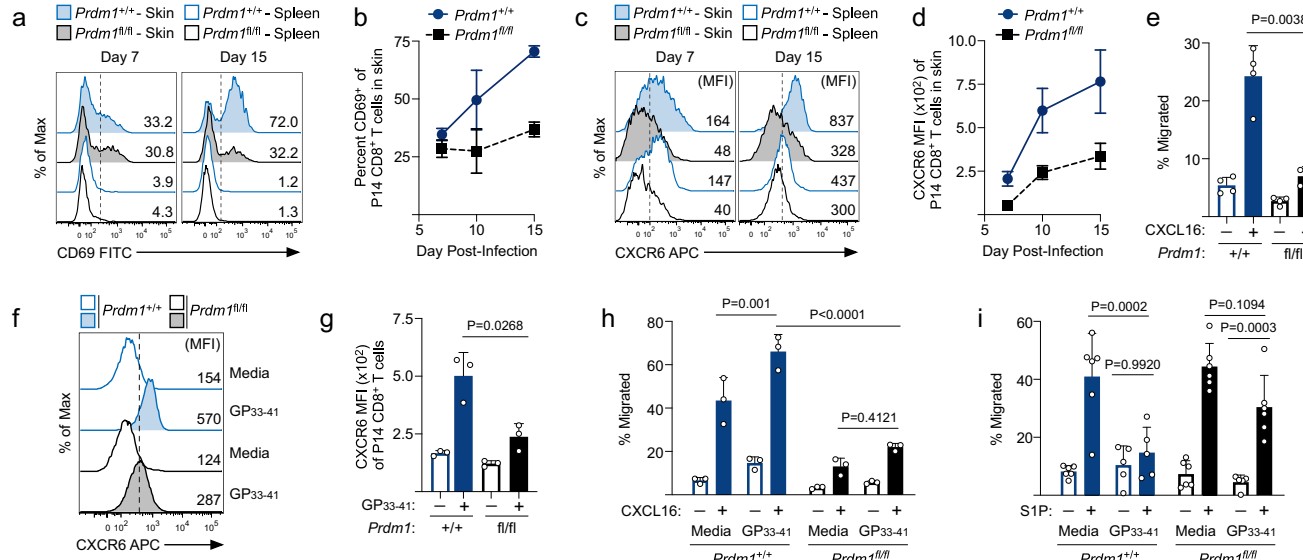

**Fig. 7 | Blimp1 is required for effector CD8⁺ T cells to change their chemotactic properties that promote tissue-residency following TCR engagement. a–i** Same experimental design shown in Fig. 6a. **a** Representative histograms depicting the expression of CD69 by *Prdm1*⁺/⁺ and *Prdm1*^fl/fl P14 CD8⁺ T cells in the skin and spleen on the indicated time points post infection. **b** Quantification of (**a**); *n* = 5, 7 and 4 for days 7, 10 and 15, respectively. **c** Same as in (**a**), but the expression of CXCR6 is displayed. **d** Quantification of (**c**); *n* = 5, 7 and 4 for days 7, 10 and 15, respectively. **e** Migration of *Prdm1*⁺/⁺ and *Prdm1*^fl/fl P14 CD8⁺ T cells isolated from the skin on day 15 post-infection in response to 10 ng/ml CXCL16; *n* = 4.
**f** Representative histograms depicting the expression of CXCR6 by *Prdm1*⁺/⁺ and

*Prdm1*^fl/fl P14 CD8⁺ T cells following ex vivo stimulation of the splenocytes with 10 nM GP₃₃-₄₁. **g** Quantification of (**f**); *n* = 3. **h, i** Same as in (**a**), but total splenocytes containing *Prdm1*⁺/⁺ and *Prdm1*^fl/fl effector P14 CD8⁺ T cells were isolated on day 10 post-infection and stimulated ex vivo with 10 nM GP₃₃-₄₁ for 48 hours at 37 °C. **h** Quantification of the migration of P14 CD8⁺ T cells following stimulation in response to 10 ng/ml CXCL16; *n* = 3. **i** Quantification of the migration of P14 CD8⁺ T cells following stimulation in response to 50 nM S1P; *n* = 6. Data shown are mean ± SD and are representative of 2 independent experiments. Statistical significance was calculated using one-way ANOVA followed by Tukey's multiple comparisons test (**e, g, h, i**). Source data are provided as a Source Data file.

CD8[+] T cells. Alternatively, Blimp1 was shown to require cooperation with its closely related homolog Hobit in regulating $T_{RM}$ differentiation of NK T cells and HSV-1-specific CD8[+] T cells, whereas Blimp1[−/−] CD8[+] T cells actually formed more $T_{RM}$ compared with WT T cells in the skin following HSV-1 infection[9]. A subsequent study found that $T_{RM}$ differentiation in the lung following influenza infection was also dependent on Blimp1, but interestingly, differentiation did not require Hobit[41]. Because $T_{RM}$ differentiation within the lung is also dependent on local antigen-recognition[22], but occurs in largely an antigen-independent manner following HSV-1 infection of the skin[42], those collective findings agree with our data presented here highlighting a critical role for TCR-stimulated expression of Blimp1 to program the chemotactic features of effector CD8[+] T cells that establish tissue-residency within nonlymphoid tissues. Our findings also demonstrate that TCR stimulation of effector CD8[+] T cells in the skin promoted the sustained expression of both CD69 and CXCR6 in a Blimp1-dependent manner. Interestingly, we showed that effector (both WT and Blimp1[−/−]) CD8[+] T cells enter the circulation highly responsive to S1P-mediated migration. These collective findings support a model that following activation, effector CD8[+] T cells enter the circulation where they poised to exit tissues following extravasation, but a secondary antigen encounter in the periphery that is sufficient to cause Blimp1 expression alters the chemotactic properties that promote retention within tissue microenvironments. Although our study here focused primarily on Blimp1-dependent changes in T cell migration, it will be of interest to also investigate whether other signaling pathways that are regulated by strength of antigen stimulation (differential expression of ICOS, for example) could also contribute to $T_{RM}$ differentiation within nonlymphoid tissues such as the skin.

Our results also demonstrate that antigen recognition by effector CD8[+] T cells heightens CXCR6 expression in a Blimp1-dependent manner, a chemokine receptor expressed by $T_{RM}$ CD8[+] T cells found in human skin and broadly implicated in $T_{RM}$ differentiation across a number of different tissues[43]. The expression of the CXCR6 ligand CXCL16 has been reported in a variety of different cell types including dendritic cells, endothelial cells, and epidermal keratinocytes[44,45]. In vivo, CXCL16 exists in both a soluble as well as a membrane-bound form and therefore has been suggested to promote not only chemotaxis, but also potentially adhesion/retention[46]. The idea that CXCR6 has evolved to exert more than just chemotactic functions has been suggested through the finding that the chemotactic activity of CXCR6 is relatively weak compared with other chemokine receptors due to the presence of an Aspartate-Arginine-Phenylalanine (DRF) motif within its transmembrane domain rather than the highly conserved Aspartate-Arginine-Tyrosine (DRY) motif found in all other chemokine receptors. In accordance with that, the maintenance of $T_{RM}$ lacking CXCR6 was impaired within a number of nonlymphoid tissues suggesting that, apart from its chemotactic role, CXCR6 may be uniquely important to establish residency possibly through the interaction with parenchymal cells expressing the membrane-bound CXCL16 within nonlymphoid tissues. Spatiotemporal expression analysis of the membrane-bound CXCL16 by various cell types within nonlymphoid tissues during and after infection, as well as their potential interaction with CXCR6 expressing T cells will need to be investigated in order to elucidate how tissue-residency of T cells is enhanced through the CXCR6-CXCL16 signaling axis.

In summary, here we used an IFNγ-YFP reporter system to identify CD8[+] T cells actively receiving TCR stimulation within the VacV-infected skin microenvironment. IFNγ[+] CD8[+] T cells exhibit phenotypic, transcriptional, and functional features of mature $T_{RM}$ T cells, suggesting that T cells actively executing effector functions comprise a significant population of $T_{RM}$ precursors. Mechanistically, TCR stimulation of effector CD8[+] T cells drove expression of the key transcription factor Blimp1, which was necessary for establishing chemotaxis properties that support $T_{RM}$ differentiation. Additionally, our data suggest that the transcriptional profile of IFNγ[+] $T_{RM}$ precursor cells may be leveraged as a resource for generating hypotheses about genes and signaling pathways involved in antigen-dependent vs. -independent $T_{RM}$ differentiation. Altogether, this study demonstrates that effector CD8[+] T cells receiving TCR stimulation are the major $T_{RM}$ precursors in the skin and defines the transcriptional pathways engaged therein, which could ultimately be utilized to optimize $T_{RM}$ formation in the context of either vaccine design or immunotherapies.

## Methods

### Ethical statement
All animal experiments were conducted in accordance with the Animal Welfare Act and the recommendations in the Guide for the Care and Use of Laboratory Animals of the National Institutes of Health. Approved by the OHSU Institutional Animal Care and Use Committee (Protocol Number IP00715) and Institutional Biosafety Committee (Registration Number IBC-13-33).

### Mice and infections
C57BL/6 mice (6–10 weeks of age, female) were purchased from Charles River/NCI (Catalog #556). IFNγ-YFP[47] (Strain #017580) and ROSA26-Cre-ER[T2][48] (Strain #008463) mice were purchased from the Jackson Laboratory. *Prdm1*-flox mice (B6.129-*Prdm1*[tm1Clme]/J, The Jackson Laboratory, Strain #008100) on the C57Bl/6 J background have been described previously[49]. C57Bl/6 P14[50] (B6.Cg-*Tcra*[tm1Mom]Tg(TcrLCMV)327Sdz/TacMmjax, MMRRC Strain #037394-JAX) and OT-I[51] (C57BL/6-Tg(TcraTcrb)1100Mjb/J, The Jackson Laboratory, Strain #003831) mice were described previously and were maintained by sibling x sibling mating. For adoptive transfers, $2.5 \times 10^4$ – $1 \times 10^5$ naïve (>95% CD44[lo]CD62L[hi]) P14 CD8[+] Thy1.1[+] T cells or OT-I CD8[+] Thy1.1[+] T cells were injected i.v. in 200 μl of PBS. VacV skin infections were performed on anesthetized mice by placing $5 \times 10^6$ PFU of virus (in 10 μl of PBS) on the ventral side of the ear pinna, then poking the virus-coated skin 25 times with a 27-gauge needle. VacV-GP33 has been previously described[52]. VacV-SIINFEKL variants were generated by homologous recombination as described previously[53] by Dr. James Gibbs in the laboratory of Dr. Jon Yewdell. All VacV strains were maintained by propagation in BSC-40 cells. Mice were euthanized by $CO_2$ asphyxiation under controlled conditions. This method of euthanasia is consistent with the American Veterinary Medical Association Guidelines for the Euthanasia of Animals. All animal experiments were approved by the OHSU Institutional Animal Care and Use Committee and Institutional Biosafety Committee.

### Leukocyte isolation from skin
Ears of infected mice were removed and the dorsal and ventral sides of the ear pinna were separated and incubated for 1.5 h at 37 °C with 1 ml HBSS (Gibco) containing $CaCl_2$ and $MgCl_2$ supplemented with 125 U/ml collagenase II (Invitrogen) and 60 U/ml DNase-I (Sigma-Aldrich). Whole-tissue suspensions were generated by gently forcing the tissue through a wire mesh screen. Leukocytes were then purified by resuspending the cells in 10 ml of 35% Percoll (GE Healthcare)/HBSS followed by centrifugation at 500 g for 10 minutes at room temperature with no brake. Cell numbers in skin were quantified by flow cytometry.

### Dermis/epidermis separation
Dermal and epidermal sections of skin were prepared by incubating ear skin in Dispase (2.5 mg/ml) for 90 minutes at 37 °C in PBS, followed by manual separation of the epidermal sheet from the dermis. Epidermal sheets were then digested in 0.25% Trypsin + 0.1% EDTA and dermal sections were digested in 125 U/ml collagenase II and 60 U/ml DNase-I. Digested dermis and epidermis were then forced through a mesh screen to generate a single cell suspension that was then stained for flow cytometry.

## Cell staining and flow cytometry

Spleens of infected mice were harvested and single-cell suspensions were generated by gently forcing the spleen through a mesh screen. Red blood cells were lysed by resuspending cell pellets in 150 mM NH4Cl, 10 mM KHCO3, and 0.1 mM Na-EDTA and staining for surface antigens was performed in PBS/1% FBS for 15 minutes at 4 °C. Data were acquired using either a BD LSRII, BD Fortessa, or a BD Symphony Flow Cytometer using BD FACSDiva version 9 in the OHSU Flow Cytometry Core Facility. Flow cytometry data were analyzed using FlowJo software, version 9.9 or 10.

## Ex vivo peptide stimulation and intracellular stain

Spleens of VacV-GP33 or VacV-SIINFEKL infected mice were harvested on the indicated day postinfection and single-cell suspensions were generated as described in Cell staining and flow cytometry section. For intracellular cytokine stain, splenocytes were seeded in a 96-well plate (2–3 million cells/well) and incubated with $GP_{33-41}$ or SIINFEKL APL variants (Biosynthesis) in the presence of 1X Brefeldin A (BioLegend) for 5 hours at 37 °C. Cells were then washed once and stained for surface antigens as described above followed by incubation with Cytofix/Cytoperm solution (BD Biosciences) for 30 minutes at 4 °C. Cells were then washed with Perm/Wash Buffer (BD Biosciences), then incubated with IFNγ antibody diluted in Perm/Wash Buffer for 30 minutes at 4 °C, washed twice in Perm/Wash Buffer and resuspended in PBS/1%FBS for analysis by flow cytometry as described above. Intracellular staining for Blimp1 or TCF-1 was performed by incubating the cells in the Transcription Factor Fix/Perm Buffer (Tonbo Biosciences) for 45 minutes at 4 °C followed by washing twice in Perm/Wash Buffer (BD Biosciences). Cells were incubated with the antibodies against Blimp1 or TCF-1 in Perm/Wash Buffer for 1 hour at 4 °C followed by washing two more times with Perm/Wash Buffer. Cells were then resuspended in PBS/1% FCS for analysis by flow cytometry. FK506 (Selleckchem) to inhibit Blimp1 expression was used at a concentration of 1 nM. To quantify IFNγ protein expression directly ex vivo, mice were injected intravenously with Brefeldin A (250 µg/mouse; B5936-Sigma) on day 5 postinfection with VacV-GP33 and were euthanized 18 hours later.

## Antibodies

The following antibodies along with appropriate isotype controls were used in this study: CD45.2 PE/Cyanin-7, Pacific Blue or APC (1:400; Clone 104; BioLegend; Cat# 109830, 109820 or 109814), CD8α Brilliant Violet 711, Pacific Blue, APC, or BUV395 (1:400; Clone 53-6.7; BioLegend; Cat# 100759, 100725, 100711, or BD Biosciences Cat# 563786), CD44 Pacific Blue (1:400; Clone IM7; BioLegend; Cat# 103020), Thy1.1 PerCP/Cy5.5, Brilliant Violet 711, or Pacific Blue (1:1000; Clone OX7; BioLegend; Cat# 109004, 202539 or 202522), Thy1.2 Brilliant Violet 605 or Pacific Blue (1:1000; clone 53-2.1; BioLegend; Cat# 140318 or 140306), KLRG1 Violet Fluor 450 (1:200; Clone 2F1; Tonbo; Cat# 75-5893-U100), CD8β PerCP/Cy5.5 (Clone YST156.7.7; BioLegend; Cat# 126609), CD69 FITC or Pacific Blue (1:100; Clone H1.2FE, BioLegend; Cat# 104506 or 104523), CD103 PE (1:200; Clone 2-E7; BioLegend; Cat# 121406), IFNγ APC (1:200; Clone XMG1.2; BioLegend; Cat# 505810), PD-1 PE (1:100; Clone 29F1.1a12; BioLegend; Cat# 135206), ICOS FITC or PE (1:200; Clone 7E.17G9, eBioscience; Cat# 11-9942-82 or 12-9942-82), CXCR6 APC (1:300; Clone SA051D1, BioLegend; Cat# 151106), Blimp1 PE (1:200; Clone 5-E7, BioLegend; Cat# 150006), TCF-1 PE (1:200; Clone S33-966; BD Bioscience; Cat# 564217), Ki-67 PE/Cyanin7 (1:200; Clone 16A8, BioLegend; Cat# 652425), CD62L PE or APC (1:400; Clone MEL-14; BioLegend; Cat# 104408 or 104412), CD25 PE (1:200; Clone PC61; BioLegend; Cat# 102007) and Viability-Ghost Dye Red 780 (1:1000; Tonbo; Cat #13-0865-T100).

## Cell sorting and microarray analysis

Mice that received IFNγ-YFP P14 CD8+ T cells were infected on the left and right ear skin with VacV-GP33. On day 7 postinfection, whole ear skin and spleens were harvested and single-cell suspensions were pooled together from 5 separate mice (10 total ears, 5 spleens). Pooled single-cell suspensions were stained for CD45, CD8, and Thy1.1 and sorted directly into TRIzol (Invitrogen) based on YFP expression using a BD InFlux sorter. Between 58,000-88,000 YFP+ cells, 100,000 YFP-, and 100,000 splenic P14 cells were collected for each replicate. RNA was isolated using chloroform-ethanol extraction followed by purification using RNeasy columns (Qiagen, Cat# 74004). Labeled cDNA was synthesized using the GeneChip Pico assay (Applied Biosystems, Cat# 902622). Amplified and labeled cDNA target samples were hybridized to an Affymetrix GeneChip Clariom S Mouse microarray (Applied Biosystems, Catalog # 902930 and image processing was performed using Affymetrix Command Console (AGCC) v3.1.1. Expression analysis was performed using Affymetrix Transcriptome Analysis Console v.4.0.3. Principal component analysis was performed on the set of differentially expressed genes between all 3 conditions using Clustergrammer[54]. Heatmap generation was obtained by performing hierarchical clustering of all differentially expressed genes using a one minus pearson correlation within the Morpheus webtool (Broad Institute, https://software.broadinstitute.org/morpheus). Gene set enrichment analysis was performed with expression data from Pan et al.[33] (GSE79805) using the java desktop application (Broad Institute).

## Quantitative PCR

Purified effector P14 CD8+ T cells were stimulated as described and RNA was isolated using an RNeasy mini kit (Qiagen, Cat# 74104) and cDNA was synthesized using the SuperScript III First Strand Kit (Invitrogen, Cat# 18080051) according to the manufacturer's instructions. qPCR reactions were performed using Power SYBR green PCR Master Mix (ThermoFisher, Cat #4368577) and analyzed on a Step One Plus Real-Time PCR system (Applied Biosystems). Changes in gene expression were quantified using the ΔΔCt method, using TATA-binding protein (*tbp*) for normalization. The following primers were used:

|  | Forward Primer | Reverse Primer |
| --- | --- | --- |
| *Ifng* | AGCAACAGCAAGGCGAAAA | GAATGCTTGGCGCTGGA |
| *S1pr1* | GTGTAGACCCAGAGTCCTGCG | AGCTTTTCCTTGGCTGGAGAG |
| *Klf2* | CTCAGCGAGCCTATCTTGCC | CACGTTGTTTAGGTCCTCATCC |
| *Ccr7* | GGGTTCCTAGTGCCTATGCTGGCTATG | GGCAATGTTGAGCTGCTTGCTGGTT |
| *Tbp* | TGGAATTGTACCGCAGCTTCA | ACTGCAGCAAATCGCTTGGG |

## Migration assays

Spleens of VacV-GP33 or VacV-SIINFEKL infected mice containing either P14 CD8+ T cells or OT-I CD8+ T cells, respectively, were harvested on day 10 postinfection and single-cell suspensions were generated as described. Cells were washed twice in RPMI supplemented with 10% FBS. Cells ($15 \times 10^6$ cells/ml) were then cultured in media alone or in media containing 10 nM $GP_{33-41}$ for 18 – 48 hours. Following peptide stimulation, cells were washed twice in RPMI 1640 supplemented with 0.5% fatty acid-free BSA (A7030, Sigma-Aldrich), L-glutamine, Penicillin/Streptomycin, 10 mM HEPES buffer, and 0.5% fatty-acid free BSA (0.5% BSA RPMI) then resuspended in the same media and left to rest for 3 hours at 37 °C. Migration assays were then performed in Transwell inserts with a pore size of 5 µM and a diameter of 6.5 mm in 24 well plates (Corning Costar). A gradient was established by plating 100 µl of cells in the upper well and 600 µl of 0.5% BSA RPMI containing the indicated concentration of S1P (Sigma), recombinant mouse CXCL16 (R & D Systems), or recombinant mouse CCL21 (R & D

Systems) in the lower well. Plates were incubated at 37 °C in 5% $CO_2$ for 3 hours and the number of cells in each well were quantified by flow cytometry.

## Tamoxifen treatment
Tamoxifen (Sigma) was dissolved aseptically in corn oil (Sigma) at a concentration of 20 mg/ml by incubation at 37 °C on a rocking plate for 18 hours. Mice received 100 µl of Tamoxifen solution i.p. one day before the transfer of naïve T cells and continued daily for 5 consecutive days (2 mg/mouse/day). Tamoxifen solutions were freshly made for each experiment.

## Statistical analysis
Data represent 2 or more independent experiment and are expressed as mean ± SD or mean ± SEM. A two-sided student's *t* test was performed to determine statistical significance between 2 groups. One-way analysis of variance (ANOVA) followed by either Tukey's or Dunnett's multiple comparisons tests were performed to determine statistical significance between multiple groups. The statistical analysis was conducted using GraphPad Prism version 9 (GraphPad Software). $P < 0.05$ was considered significant for all statistical analysis.

## Reporting summary
Further information on research design is available in the Nature Portfolio Reporting Summary linked to this article.

## Data availability
The gene expression analysis data generated by Affymetrix GeneChip Clariom S Mouse microarray have been deposited in the NCBI Gene Expression Omibus (GEO) Database under accession code GSE233935. All other data are available within the article, Supplementary Information or Source data file or from the author upon request. Source data are provided with this paper.

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

## Acknowledgements

The authors would like to thank Jake Harbour, Samantha Fancher, Jack McLean, and Taylen Nappi for helpful discussion and assistance. Flow cytometry and cell sorting were performed in the OHSU Flow Cytometry Core Facility and microarray assays were performed in the OHSU Gene Profiling Shared Resource. This work was supported by the National Institute of Health grants R01-AI132404 and R01-AI143664 to JCN.

## Author contributions

M.A., S.J.H., and J.C.N. designed and performed the experiments, analyzed the data and wrote the manuscript. J.S.G. and J.W.Y. generated and provided critical reagents.

## Competing interests

The authors declare no competing interests.
