## [Peer Review File · Nature Communications]

T cell receptor signaling strength establishes the chemotactic properties of effector CD8⁺ T cells that control tissue-residencyREVIEWER COMMENTS

Reviewer #1 (Remarks to the Author):

An IFN γ -YFP reporter system has been used to examine the requirements for TRM development in the skin. The data show that secondary TCR stimulation induces Blimp 1 expression and triggers a chemotactic switch that supports residency in the local tissues. Since the mechanisms that control the functional properties of TRM cells are poorly understood, the study is timely and significant. Although the study is interesting, some important questions need to be addressed.

1) Some aspects of the data presentation and discussion are confusing. Although CD69 and CD103 are the predominant markers of TRM cells in cutaneous and mucosal tissues, co-expression is only shown in figure 6. Prior studies show that naïve and central memory CD8 T cells both generate TRM cells after antigen stimulation. After activation, naïve CD8 T cells upregulate CD103 in the presence of TGF β , whereas TCM cells are refractory to cytokine stimulation and produce TRM cells that express CD69 without CD103 (Osborne et al Plos Pathogens 2019). The reason for this difference has not been defined, but may reflect epigenetic-modifications that are introduced after antigen-stimulation. Since the goal of the current study is to define how TCR stimulation influences TRM development, these markers should be analyzed simultaneously. It would be helpful to know whether IFN γ + TRM cells are refractory to TGF β in vitro. Since CD103 is negatively regulated by TcR stimulation (ref 24) there may be a negative correlation with IFN γ production. Although antigen plays a role in TRM development, the requirement for persisting antigen during the maintenance phase remains controversial. What was the phenotype of P14 CTLs in the right ear of the co-infected mice (figure 1,K)? How long did the cells persist?

2) Line 146 "IFN γ expressing CD8+ T cells exhibited low surface expression of KLRG1 suggesting phenotypic reprogramming of effector T cells within the inflamed skin microenvironment". The logic behind this statement is unclear. Figure "I" appears to show IFN γ + cells as percentages of the total, rather than in KLRG1+/- subsets. According to the attached contour plot, approximately 40% of KLRG+ CTLs expressed IFN γ . Data from other studies suggest that enriched populations of KLRG1+ CTLs reside in the vasculature, consequently low KLRG1 expression may reflect selective migration rather than reprogramming.

3) Line 219. The authors conclude that the strength of TCR stimulation required for IFN γ expression is largely equivalent to that required for TRM CD8+ T cell differentiation. However, there is no clear strategy for distinguishing newly activated TEFF cells from bona fide TRM cells, as both subsets express CD69 and PD-1. The IFN γ + CTLs lack Hobit expression (line 3003) and S1PR5 was not analyzed. Can the authors exclude the possibility that strong antigen stimulation retains TEFF cells in the skin, while weak antigen stimulation permits re-entry into the circulation? Do IFN γ + CTLs proliferate in the skin?

4) YFP expression is used as surrogate for TCR stimulation, however it is unclear when the TRM cells arrived in the skin. What is the half-life of YFP? Do CTLs in the skin express Nur77. IV staining provides a snap-shot during cell trafficking, but does not reveal the duration of residence in a specific tissue. Has the YFP reporter been used to analyze TRM cells after infections with other pathogens? Do the results of this study apply to other infections?

7) Line 118. The authors state that "effector CD8+ T cells actively engaging in TCR-dependent effector functions are the major TRM precursors". This statement is not fully supported by the data. Although some precursors of TRM cells have been detected within TEFF populations (Kok et JEM 2020), conversion was not shown. The data suggested that imprinting occurred before TRM cells arrived in the skin.

5) Line 844. 7dpi infection is too early to analyze "mature" TRM cells. Additional time points should be shown. How does the transcriptional profile of IFN γ + CTLs compare with peptide stimulated effector cells from the spleen? Figure 3F is not very informative, which genes are upregulated? These limited

data sets are not sufficient to draw broad conclusions about TRM precursors.

6) The function of TGFb during TRM development is not clearly discussed. A recent report found that some naïve CD8 T cells were preconditioned to become TRM cells during interactions APCs that activate TGFb (Mani et al. Science 2019). S1PR5 is required for CTLs to leave peripheral tissues and is down regulated by TGFb. What is the role of TGFb in this model?

8) Line 867 the source of effector cells used for migration experiments is not clearly indicated. When analyzed after antigen stimulation, it seems that splenic effector cells are similar to skin TRM cells, and that environmental cues play little role in genetic programming. Please clarify.

9) Line 370. The authors state that Blimp1^{-/-} CD8⁺ T cells expressed higher levels of both CD62L and TCF-1 than WT T cells (Fig 6B-E), consistent with previous reports that Blimp1 promotes terminal effector T cell differentiation in the circulation. This statement is confusing. CD62L is generally considered to be a marker of naïve and TCM cells, whereas KLRG1 is used to identify terminally - differentiated effector CD8 T cells.

10) Line 379. The observation that Blimp^{-/-} CTLs do not upregulate CD69 in the presence of antigen is interesting. Do Blimp^{-/-} CTLs express CD69 in the co-infection model (right ear – no antigen)? CD103 is down regulated by TCR and induced by TGFb, while the mechanism induces co-expression (CD103 plus CD69) is unknown.

Overall, the figures are well-presented and methods are well described.

This report was prepared by Linda Cauley.

Reviewer #2 (Remarks to the Author):

Using an elegant experimental setup, Abdelbary et al. have shown that antigen recognition in the skin significantly affects the number of organ-bound and presumably resident T cells. The surprising thing is that this is independent of the number of cells in the spleen (Fig. 2D-I). Accordingly, the authors have identified a previously unknown tissue-limited affinity-based checkpoint that influences the composition of the Trm population.

Nevertheless, there are several issues that need to be addressed in a revised manuscript.

Specific points:

> One major issue I have with the manuscript is that it needs to be better focused. It starts with the abstract, where I miss a clearer line. I mean, the manuscript has very nice insights, but I think the abstract doesn't convey them. Instead, it touches on many things while leaving the overall picture rather unclear. For me, the most important finding is that the strength of antigen recognition determines the number of resident cells independent of the number of circulating cells. I think this should be presented directly in the abstract, and then the possible mechanisms of how this happens (e.g., via control of chemotaxis) should be explained.

> It would be very informative, if the authors would not only perform NGS analysis of high antigen exposed IFNg⁺ versus IFNg⁻ cells but also when cells were exposed to different TCR strength. I mean the main message is how signal strength impacts Trm formation but surprisingly this aspect was not evaluated in the NGS analysis.

> The authors used APL expressing VV to cause infection in the skin. It would be very interesting to see what happens to Trm when other types of APL expressing Trm are tested, e.g., VSV-V4 versus

VSV-N4-containing pathogens.

> While the data shown in Figure 2 for the APL are really interesting, the follow up shown in Figure 5K-M is trivial. The authors expose effector cells to different signal strengths and see that CD69 - the key marker for recent T cell activation - is upregulated. Whether this CD69 has anything to do with T cell residency remains highly debatable. Similar arguments could be made for Blimp-1 and, to some extent, to S1p migration differences the authors have observed. Moreover, these tests were performed with total effector T cells, for which it remains unclear whether they still have the potential to form true Trm.

> The authors conclude that differences in chemokine-driven migration are critical for increasing Trm numbers upon high-affinity tissues stimulation. This conclusion is based on a series of in vitro experiments in which Blimp1, CXCR6, and CD69 were associated with TCR signaling strength. However, the MS currently lacks direct evidence that the in vivo Trm phenotype following high or low affinity stimulation is indeed related to these chemotactic pathways. I say this because it could just as easily be that lower TCR signaling strength leads to lower antigen-driven terminal expansion of Trm in tissues. Here I return to the lack of NGS analysis mentioned above, i.e., comparing the response of high and low affinity Trm in tissues. This would allow to determine whether chemokine-induced signaling or differences in proliferation lead to this phenotype of Trm signaling strength.

> The authors claim that IFN γ reporter identified all cells that were exposed to antigen but this claim is incompletely sustained. I mean the authors show that IFN γ secretion requires local provision of antigen but there is no proof that all antigen exposed cells are IFN γ positive. In fact, there might be many antigen-exposed cells that are not reporter positive.

Minor points:

The correct nomenclature of the mouse MHC is H-2D instead of H2-D.

I talked to dermatologists, but the term parenchyma is not normally used in the context of skin. The term should therefore be replaced.

In the introduction, the authors make several statements that should be changed. Line 43: T cells are not activated only in draining lymph nodes (e.g., spleen, tertiary lymphoid structures...). Line 44-46: It is not proven that inflammation must subside before Trm are formed. Line 64: The authors talk about the main regulator of Trm, but did not define what they consider as the main regulator.

In general, the introduction is long and it takes quite a while to become clear what issues have been addressed.

It is difficult to understand what the authors are trying to say in the sentences in lines 98 to 103.

Reviewer #3 (Remarks to the Author):

The molecular mechanisms regulating the formation and maintenance of tissue-resident memory T cells (Trm) are still incompletely understood. Specifically, it is not known how a second encounter with antigen in the tissue after the initial priming in lymph node can influence T cell tissue residence. The Nolz lab has previously shown that antigen in skin facilitates tissue-resident memory T cell formation during vaccinia infection. In this follow-up paper, Abdelbary et al. presents further evidence showing the critical role for tissue antigen in facilitating Trm formation. Critically, the authors demonstrate multiple mechanisms by which tissue antigen enable Tm formation: re-encounter with antigen upregulates Blimp1 transcription factor, which is required for changes in T cell chemotactic properties to promote tissue residency and for Trm residency. The authors also suggest that T cell receptor

signal strength regulates T cell chemotactic properties, with low affinity antigen being sufficient to induce migration towards CXCR6, but with strong antigens required for suppression of S1P egress. These are important findings in rapidly developing field in immunology. The experiments are very well designed, and the manuscript is clearly written.

However, there are few questions/concerns that should be address by the authors:

1. Use of IFN γ -YFP reporter: The authors present very convincing data that % of YFP+ antigen-specific T cells correlates with % of IFN-g+ cells. However, this is only shown for T cells from spleen after in vitro peptide re-stimulation. Are YFP+ T cells from skin also IFN γ +, both ex vivo and with or without antigen re-stimulation? How long can the YFP signal persist after cessation of antigen encounter? This is important for interpretation of the IFN γ -YFP model.
2. Figure 1H and I. Only KLRG1^{low} cells express in this experimental system. How does this relate to previous work, showing that both KLRG1^{low} and KLRG1^{hi} T cells can produce IFN-gamma (for example: Robbins et al. (2003) Differential Regulation of Killer Cell Lectin-Like Receptor G1 Expression on T Cells). Are KLRG1^{low} T cells from skin also CD127^{hi}? Can KLRG1^{low} cells from skin produce IFN-gamma after antigen re-stimulation in vitro? Does IFN-gamma production differ between KLRG1^{low} and KLRG1^{hi} populations from skin and from spleen?
3. Figure 1N. What is the time point for the analysis? The figure legend suggests it is day 7, but this does not agree with the conclusion on Trm formation after viral clearance.
4. Figure 5J-M: What is the time point for analysis of Blimp1 upregulation? Naïve T cells have been reported to express Blimp1 after 48h of stimulation (Gong and Malek, 2007 Cytokine-Dependent Blimp-1 Expression in Activated T Cells Inhibits IL-2 Production), so the difference observed by the authors could be due to the different kinetics of Blimp1 expression between naïve and pre-activated T cells.
5. Blimp1 conditional deletion – in the main text, the Blimp1^{-/-} mice are only introduced as Prdm1f/fl. These must have been Cre-ERT2+, as this Cre is mentioned in the Material and Methods, and mice are treated with tamoxifen. Please mention the Cre in the main text. What exactly were the Blimp1 WT controls used in these experiments – were they also Cre+ but Prdmwt/wt; where they also tamoxifen-treated? Please add data confirming the efficiency of deletion after tamoxifen treatment.
6. P14 and OTI T cells used for the adoptive transfer experiments – were the T cells sorted for naïve (CD44^{low}, CD62L^{hi}) phenotype? If not, what was the % of the naïve phenotype cells used for adoptive transfers.
7. Information on number of independent experiments using biological/technical replicates should be included in the figure legends. The means/individual data points are from single representative experiment, or from data pooled from multiple experiments?
8. In some cases (for example: 2F-1), t-tests seem to be used compare more than 2 means. Please either change this to more appropriate analysis or explain how the data was analysed.

REVIEWER COMMENTS

Reviewer #1 (Remarks to the Author):

An IFN γ -YFP reporter system has been used to examine the requirements for TRM development in the skin. The data show that secondary TCR stimulation induces Blimp 1 expression and triggers a chemotactic switch that supports residency in the local tissues. Since the mechanisms that control the functional properties of TRM cells are poorly understood, the study is timely and significant. Although the study is interesting, some important questions need to be addressed.

1) Some aspects of the data presentation and discussion are confusing. Although CD69 and CD103 are the predominant markers of TRM cells in cutaneous and mucosal tissues, co-expression is only shown in figure 6. Prior studies show that naïve and central memory CD8 T cells both generate TRM cells after antigen stimulation. After activation, naïve CD8 T cells upregulate CD103 in the presence of TGF β , whereas TCM cells are refractory to cytokine stimulation and produce TRM cells that express CD69 without CD103 (Osborne et al Plos Pathogens 2019). The reason for this difference has not been defined, but may reflect epigenetic-modifications that are introduced after antigen-stimulation. Since the goal of the current study is to define how TCR stimulation influences TRM development, these markers should be analyzed simultaneously. It would be helpful to know whether IFN γ + TRM cells are refractory to TGF β in vitro. Since CD103 is negatively regulated by TcR stimulation (ref 24) there may be a negative correlation with IFN γ production. Although antigen plays a role in TRM development, the requirement for persisting antigen during the maintenance phase remains controversial. What was the phenotype of P14 CTLs in the right ear of the co-infected mice (figure 1,K)? How long did the cells persist?

We have previously reported and analyzed extensively the co-expression of CD69 and CD103 in the skin during VacV infection and whether local recognition of antigen regulated expression of either of these T_{RM} markers (Khan et al, JEM, 2016). As the reviewer correctly indicates, TCR stimulation inhibits TGF- β -mediated expression of CD103 and we have confirmed that finding in our laboratory on multiple occasions. As reported previously in the JEM paper, we consistently find that CD103 becomes expressed ONLY following the resolution of the viral infection (VacV cannot be detected in the skin after day 15 post-infection) and this does NOT require cognate antigen recognition (this was also shown in Figure 6I,6J of this manuscript). Thus, our interpretation of these earlier findings agrees that antigen recognition likely does limit CD103 expression in the skin microenvironment and only becomes expressed once antigen has been cleared. The collective findings in this manuscript also completely agree with that interpretation. For example, IFN γ expression (YFP) cannot be detected in the skin after day 10, suggesting the antigen level in the skin after that timepoint is not high enough to cause local effector functions of CD8⁺ T cells (which we also predict would prevent CD103 expression) and CD103 only becomes expressed once antigen load has been reduced/eliminated. For clarity, in Figure 1B, we indicate the time frame that VacV can be detected in the skin based on our previously published results (Khan et al, JEM, 2016; Osborn et al, PLOS Pathogens, 2019).

In addition, as pointed out by R3, the time point for Fig 1N in the original manuscript was not clearly stated in the Figure Legend (it was day 40). We have independently verified and replicated these previously reported findings and can be found in Figure 1P-R of the revised manuscript. We have also clearly indicated the time points for individual panels of Figure 1 to assist readers in the interpretation of the data.

2) Line 146 "IFN γ expressing CD8⁺ T cells exhibited low surface expression of KLRG1 suggesting phenotypic reprogramming of effector T cells within the inflamed skin microenvironment". The logic

behind this statement is unclear. Figure “I” appears to show IFN γ + cells as percentages of the total, rather than in KLRG1+/- subsets. According to the attached contour plot, approximately 40% of KLRG1+ CTLs expressed IFN γ . Data from other studies suggest that enriched populations of KLRG1+ CTLs reside in the vasculature, consequently low KLRG1 expression may reflect selective migration rather than reprogramming.

First, we must apologize as there was a mistake in the previous Figure 1I. There was a “5” in the lower, right quadrant that was supposed to be on the axis (as part of the label “10⁵”), and was not the percentage for that quadrant. Second, we very much appreciate the comment about prior studies suggesting that KLRG1+ CD8+ T cells largely being associated with the vasculature rather than being tissue-infiltrating. To determine whether local antigen recognition changed KLRG1 expression on effector CD8+ T cells, we performed the experiment shown in Figure 1L-O. These data show that KLRG1- CD8+ T cells are enriched within the skin compared to the spleen and is NOT regulated by local antigen recognition. Thus, as the reviewer suggests, this observation is likely because of selective migration, rather than downregulation of KLRG1 during antigen recognition. These new data are found in Figure 1L-O and we have modified the text within the manuscript accordingly (lines 151-155).

3) Line 219. The authors conclude that the strength of TCR stimulation required for IFN γ expression is largely equivalent to that required for TRM CD8+ T cell differentiation. However, there is no clear strategy for distinguishing newly activated TEFF cells from bona fide TRM cells, as both subsets express CD69 and PD-1. The IFN γ + CTLs lack Hobit expression (line 3003) and S1PR5 was not analyzed. Can the authors exclude the possibility that strong antigen stimulation retains TEFF cells in the skin, while weak antigen stimulation permits re-entry into the circulation? Do IFN γ + CTLs proliferate in the skin?

We believe there is some confusion regarding IFN γ expression by CD8+ T cells in the skin. As shown in Figure 1B, IFN γ expression is ONLY detected in skin during infection before day 10 (which coincides with viral clearance). Indeed, we completely agree with the reviewers’ interpretation of the data (which is also the title of our manuscript): that strong TCR stimulation is necessary to retain the effector CD8+ T cells in the skin (by promoting CXCR6 expression and limiting S1P1-dependent egress), thereby subsequently retaining them to transition into tissue-residents after clearance of the viral infection. We have modified the text accordingly for clarity.

In our previous report (Khan et al, JEM, 2016), we used BrdU incorporation during different windows of the VacV skin infection and found no evidence that the presence of antigen in the skin promotes significant secondary proliferation of effector CD8+ T cells (see Fig 7A, B of that manuscript). To extend that finding, we also analyzed Ki67 expression in the skin and again, found no difference in expression of this proliferation marker on days 7 or 15 post-infection. These data are included in (Fig S1F,G) of the revised manuscript.

It has been previously reported that expression of Hobit cannot be readily detected in T_{RM} CD8+ T cells in the skin until ~30 days after HSV-1 infection (MacKay et al, Science, 2016) and we have found no evidence that expression of Hobit is regulated downstream of the TCR. According to our gene expression analysis, *S1pr5* expression was elevated in the skin compared to the spleen, but was not differentially expressed between IFN γ + and IFN γ - T cells (Figure S3 and data not shown).

4) YFP expression is used as surrogate for TCR stimulation, however it is unclear when the TRM cells arrived in the skin. What is the half-life of YFP? Do CTLs in the skin express Nur77. IV staining provides a snap-shot during cell trafficking, but does not reveal the duration of residence in a specific

tissue. Has the YFP reporter been used to analyze TRM cells after infections with other pathogens? Do the results of this study apply to other infections?

We again would like to reiterate that T_{RM} CD8⁺ T cells in the skin DO NOT continue to produce IFN_γ following the resolution of VacV infection. The kinetics of YFP expression are shown in Fig 1B of the manuscript and show that expression of IFN_γ cannot be readily detected after day 10 post-infection. Thus, our collective findings clearly demonstrate that antigen recognition that is necessary for IFN_γ expression during this window has a profound influence on downstream gene expression (Fig 3 and S3) that ultimately promotes retention in the skin and T_{RM} differentiation.

To estimate the half-life of YFP, we stimulated effector P14 CD8⁺ T cells from the spleen overnight with GP₃₃₋₄₁ peptide, then washed the T cells extensively and monitored loss of YFP expression over time. These data are now shown in Fig S1C and suggest that the half-life of YFP using our experimental approach is approximately 14.2 hours.

Our gene expression analysis shows that IFN_γ⁺ T cells in the skin are also expressing Nur77 (*Nr4a1*). This was indicated in the heat map shown in Figure S3C. We also measured Nur77 expression previously and found increased expression in Ag-rich skin (Khan et al, JEM, 2016).

The IV labeling experiment shown in Fig 1D,E was a control experiment to clearly demonstrate that the IFN_γ producing T cells (on day 7) had clearly infiltrated the skin and were not associated with the vasculature. This is an important control experiment as it demonstrates 1) that all IFN_γ expressing T cells had infiltrated the dermis/epidermis and 2) not all T cells that enter the skin produce IFN_γ.

To our knowledge, we do not believe our novel strategy of using an IFN_γ reporter to identify T cells actively executing effector function in vivo has been used in any other studies to identify the mechanisms that promote tissue-residency and the utility and robustness of this strategy may differ between different types of infections and likely reflects access to and the amount of antigen presentation within a given tissue. For example, T_{RM} differentiation is known to be regulated by antigen recognition in some tissues (skin, lung, brain), thus we would predict that during those types of infections (influenza infection of the lung, for example), effector CD8⁺ T cells would produce IFN_γ in an antigen-dependent manner and this would be directly linked to transcriptional changes and chemotactic shifts that promote tissue residency, as we have described throughout our manuscript.

7) Line 118. The authors state that “effector CD8⁺ T cells actively engaging in TCR-dependent effector functions are the major TRM precursors”. This statement is not fully supported by the data. Although some precursors of TRM cells have been detected within TEFF populations (Kok et JEM 2020), conversion was not shown. The data suggested that imprinting occurred before TRM cells arrived in the skin.

We have removed this statement from the manuscript.

5) Line 844. 7dpi infection is too early to analyze “mature” TRM cells. Additional time points should be shown. How does the transcriptional profile of IFN_γ⁺ CTLs compare with peptide stimulated effector cells from the spleen? Figure 3F is not very informative, which genes are upregulated? These limited data sets are not sufficient to draw broad conclusions about TRM precursors.

We have modified the figure legend accordingly (now lines 813-815).

One of the major goals of this study was to identify the changes in gene expression that occur in CD8⁺ T cells once they encounter antigen within non-lymphoid tissues during infection and whether those changes in gene expression were consistent with those T cells acquiring a tissue-residency signature. Gene set enrichment analysis (GSEA) is a commonly used bioinformatic analysis to determine whether the overall changes in gene expression would indicate a 'biological process'. For the analysis shown in Fig 3F and 3G, we used previously published expression data directly comparing T_{RM} CD8⁺ T cells and T_{EM} CD8⁺ T cells that form after VacV skin infection as our defined gene set and "tested" whether the differentially expressed genes between IFN γ ⁺ and IFN γ ⁻ T cells were more enriched within either of these subsets. Strikingly, these data clearly indicate (with a highly significant FDR $q < 0.001$) that IFN γ ⁺ T cells express genes strongly associated with T_{RM} CD8⁺ T cells and suppress genes that more closely resemble T_{EM} CD8⁺ T cells. A number of these genes are highlighted in Fig S3C. In addition, there are clearly environmental factors within the skin that also cause changes in gene expression (see Fig 3D,E, S3B, S3C). For the genes we identified as being dependent on TCR signaling, we confirmed those findings extensively (Fig 4, 5, S5). Thus, we do not see the value in performing a complete gene expression profile on peptide stimulated CD8⁺ T cells from the spleen. Although any individual study on its own may not allow for "broad conclusions", the data are entirely consistent with the interpretation that antigen recognition during infection is promoting expression of a gene signature that closely resembles that of mature T_{RM} CD8⁺ T cells.

6) The function of TGF β during TRM development is not clearly discussed. A recent report found that some naïve CD8 T cells were preconditioned to become TRM cells during interactions APCs that activate TGF β (Mani et al. Science 2019). S1PR5 is required for CTLs to leave peripheral tissues and is down regulated by TGF β . What is the role of TGF β in this model?

Shortly after the manuscript by Mani et al was published in 2019, we performed the same sorting experiment and analyzed T_{RM} CD8⁺ T cell differentiation following a VacV skin infection. We found that CD8⁺ T cells in the skin expressed similar levels of CD103 regardless if they originated from CD103⁺ or CD103⁻ naïve T cells and generated similar quantities of T_{RM} CD8⁺ T cells following VacV skin infection.

During VacV infection of the skin, CD103 only becomes expressed by CD8⁺ T cells after detectable virus has been cleared from the skin. This occurs consistently between days 10 and 15 post-infection and has been previously reported by us (Khan et al, JEM, 2016; Osborn et al, Plos Pathogens, 2019). Additionally, we performed an experiment (Fig S7A,B) showing similar upregulation of CD103 expression by WT and Blimp1^{-/-} CD8⁺ T cells following in vitro incubation with TGF- β for 48 hours, consistent with similar levels of CD103 expression by WT and Blimp1^{-/-} CD8 T cells within the skin on day 15 post VacV infection (Fig 6I) indicating that CD103 expression does not require antigen recognition or Blimp1.

In our gene expression analysis, we found no difference in the expression of S1PR5 between the YFP⁺ and YFP⁻ T cells. Although we cannot conclusively rule out a role for S1PR5 in promoting tissue residency in the skin, its expression does not seem to be regulated downstream of the TCR, which was the goal of this study.

8) Line 867 the source of effector cells used for migration experiments is not clearly indicated. When analyzed after antigen stimulation, it seems that splenic effector cells are similar to skin TRM cells, and that environmental cues play little role in genetic programming. Please clarify.

We have clarified that for the migration assays in Fig 4, the effector T cells were from the spleen. As mentioned previously, there are clearly environmental factors within the skin that also cause changes in gene expression (see Fig 3D,E, S3B, S3C). However, with regards to altering chemotaxis, it seems that antigen recognition alone is the major factor. In addition, our data in Fig1Q,R show that some T_{RM} CD8⁺ T cells will form in an antigen-independent manner, although this process is highly inefficient, but nevertheless, demonstrates the importance of environmental cues to also promote T_{RM} differentiation.

9) Line 370. The authors state that Blimp1^{-/-} CD8⁺ T cells expressed higher levels of both CD62L and TCF-1 than WT T cells (Fig 6B-E), consistent with previous reports that Blimp1 promotes terminal effector T cell differentiation in the circulation. This statement is confusing. CD62L is generally considered to be a marker of naïve and TCM cells, whereas KLRG1 is used to identify terminally - differentiated effector CD8 T cells.

As reported previously, Blimp1^{-/-} effector/memory CD8⁺ T cells express lower markers of terminal effector differentiation and higher levels of central memory markers. Thus, we used expression of CD62L and TCF-1 to indicate sufficient deletion of Blimp1 in the circulating effector T cells. We have modified this statement for clarity (lines 367-369).

10) Line 379. The observation that Blimp^{-/-} CTLs do not upregulate CD69 in the presence of antigen is interesting. Do Blimp^{-/-} CTLs express CD69 in the co-infection model (right ear – no antigen)? CD103 is down regulated by TCR and induced by TGF β , while the mechanism induces co-expression (CD103 plus CD69) is unknown.

These data were already included in the manuscript (Figure 6). Expression of CD69 requires both antigen and Blimp1, whereas expression of CD103 does not require antigen or Blimp1.

Overall, the figures are well-presented and methods are well described.

This report was prepared by Linda Cauley.

Reviewer #2 (Remarks to the Author):

Using an elegant experimental setup, Abdelbary et al. have shown that antigen recognition in the skin significantly affects the number of organ-bound and presumably resident T cells. The surprising thing is that this is independent of the number of cells in the spleen (Fig. 2D-I). Accordingly, the authors have identified a previously unknown tissue-limited affinity-based checkpoint that influences the composition of the T_{RM} population.

Nevertheless, there are several issues that need to be addressed in a revised manuscript.

Specific points:

> One major issue I have with the manuscript is that it needs to be better focused. It starts with the abstract, where I miss a clearer line. I mean, the manuscript has very nice insights, but I think the abstract doesn't convey them. Instead, it touches on many things while leaving the overall picture rather unclear. For me, the most important finding is that the strength of antigen recognition determines the number of resident cells independent of the number of circulating cells. I think this should be presented

directly in the abstract, and then the possible mechanisms of how this happens (e.g., via control of chemotaxis) should be explained.

We appreciate that the reviewer finds the data using the VacV expressing altered peptide ligands of SIINFEKL so interesting, yet, we respectfully disagree that this is the main message of the manuscript. The primary message of the manuscript is that level of TCR stimulation that is necessary to stimulate effector functions (i.e., IFN γ production) causes profound changes in gene transcription that promote tissue-residency, in particular, by altering the chemotactic properties of effector CD8 $^+$ T cells that is dependent on expression of Blimp1 following secondary antigen encounter. We believe the abstract accurately and succinctly reflects this message.

> It would be very informative, if the authors would not only perform NGS analysis of high antigen exposed IFN γ $^+$ versus IFN γ $^-$ cells but also when cells were exposed to different TCR strength. I mean the main message is how signal strength impacts T_{RM} formation but surprisingly this aspect was not evaluated in the NGS analysis.

The series of experiments performed in Figures 1 and 2 (along with the associated supplemental figures) were used to clearly establish the utility of the IFN γ reporter system to identify CD8 $^+$ T cells that are receiving sufficient TCR stimulation to execute effector functions. As we show in Fig 2A, different concentrations of SIINFEKL variants are necessary to promote IFN γ production by effector CD8 $^+$ T cells and this was then confirmed in vivo using newly engineered Vaccinia viruses expressing the SIINFEKL variants. Thus, if our prediction that a certain threshold of TCR stimulation was necessary to stimulate IFN γ production, then fewer tissue-residents would form within skin where lower affinity peptide ligands were present and that the IFN γ -expressing T cells were the key subset undergoing major changes in gene expression leading to tissue-residency. Indeed, this was the case. These important results fully supported our main hypothesis that we could identify antigen-specific, IFN γ $^+$ CD8 $^+$ T cells in the skin and that these T cells were likely the major T_{RM} precursors. To then test that hypothesis directly, we sorted IFN γ $^+$ and IFN γ $^-$ T cells from the same skin microenvironment and found dramatic changes in gene expression between those two subsets. This allowed us to identify critical targets of TCR signaling following a second antigen encounter that we then show regulate T_{RM} differentiation.

We would also like to address the technical feasibility of the suggested analysis. Sorting antigen-specific T cells from ear skin after VacV infection is not trivial and took many animals to acquire enough mRNA to perform the transcriptional analysis shown in Figure 3 and Figure S3. Sorting the necessary number of effector OT-I CD8 $^+$ T cells from skin infected with each of the APLs along with the VacV-only control would require (at minimum) 150 animals, not to mention the many hours of sorting. We also believe that the results from this costly and time-consuming experiment are highly predictable and would simply reflect the percentage of T cells that are receiving sufficient TCR signaling to cause expression of IFN γ , Blimp1, ICOS, CD69, CXCR6, etc., and suppress Klf2, S1pr1, etc., which we already demonstrate in Figure 3, 4 and 5, as well as throughout our manuscript.

> The authors used APL expressing VV to cause infection in the skin. It would be very interesting to see what happens to T_{RM} when other types of APL expressing T_{RM} are tested, e.g., VSV-V4 versus VSV-N4-containing pathogens.

One powerful feature of the VacV skin infection model that we have characterized extensively in our laboratory is that the virus is unable to spread systemically. Thus, we are able to co-infect mice with two “different” VacV expressing model antigens or lower affinity variants to directly test the role for TCR signaling strength within two microenvironments all within the same animal. As we clearly point out in Supplemental Figure 2, activation and expansion of OT-I CD8⁺ T cells is dramatically lower following infection with VacV expressing Q4, T4, or V4 SIINFEKL variants. In order to experimentally assess whether strength of secondary antigen stimulation within non-lymphoid tissues regulated tissue-residency, we needed to co-infect with VacV-N4 on the same animal (Figure 2), so that a measurable, common pool of “effector T cells” are generated and recruited into each site of infection equally. This approach allowed us to determine in a highly controlled experimental setting the transcriptional changes that occur following a secondary antigen encounter within non-lymphoid tissues such as the skin and whether it promotes the differentiation of tissue-resident T cells.

When using broadly systemic infections such as VSV, limiting presentation of an antigen to a defined tissue microenvironment is not possible. TCR signal strength regulates many aspects of CD8⁺ T cell activation, proliferative expansion, and differentiation (as we show in Supplemental Figure 2 and discussed above). Thus, that type of viral infection model would not be able to address the specific question we are asking here, which is how does TCR signal strength within tissue microenvironments control the extent of T_{RM} differentiation.

> While the data shown in Figure 2 for the APL are really interesting, the follow up shown in Figure 5K-M is trivial. The authors expose effector cells to different signal strengths and see that CD69 - the key marker for recent T cell activation - is upregulated. Whether this CD69 has anything to do with T cell residency remains highly debatable. Similar arguments could be made for Blimp-1 and, to some extent, to S1p migration differences the authors have observed. Moreover, these tests were performed with total effector T cells, for which it remains unclear whether they still have the potential to form true T_{RM}.

We respectfully disagree that the data shown in Figure 5 are trivial. The goal of that series of experiments was to determine whether antigen recognition by effector CD8⁺ T cells would be sufficient to cause Blimp1 to be expressed and for the chemotactic properties of effector CD8⁺ T cells to be altered. Importantly, these data convincingly demonstrate that similar TCR signal strength is required for Blimp1 expression and to suppress S1P migration that regulates tissue egress. These are critical findings directly linked to the mechanisms that ultimately establish T_{RM} T cells within the skin and perhaps other non-lymphoid tissues.

> The authors conclude that differences in chemokine-driven migration are critical for increasing T_{RM} numbers upon high-affinity tissues stimulation. This conclusion is based on a series of in vitro experiments in which Blimp1, CXCR6, and CD69 were associated with TCR signaling strength. However, the MS currently lacks direct evidence that the in vivo T_{RM} phenotype following high or low affinity stimulation is indeed related to these chemotactic pathways. I say this because it could just as easily be that lower TCR signaling strength leads to lower antigen-driven terminal expansion of T_{RM} in tissues. Here I return to the lack of NGS analysis mentioned above, i.e., comparing the response of high and low affinity T_{RM} in tissues. This would allow to determine whether chemokine-induced signaling or differences in proliferation lead to this phenotype of T_{RM} signaling strength.

In our previous report (Khan et al, JEM, 2016), we used BrdU incorporation during different windows of the infection and found no evidence that antigen in the skin causes secondary local proliferation of effector CD8⁺ T cells (see Figure 6A-C of that manuscript). To extend that finding, we have also analyzed Ki67 expression in the skin and again, found no difference in expression of this proliferation marker on either day 7 or 15 post-infection (+/- antigen all on the

same animal). The Ki67 expression data is included in (Fig S1F,G) of the revised manuscript.

> The authors claim that IFN γ reporter identified all cells that were exposed to antigen but this claim is incompletely sustained. I mean the authors show that IFN γ secretion requires local provision of antigen but there is no proof that all antigen exposed cells are IFN γ positive. In fact, there might be many antigen-exposed cells that are not reporter positive.

We performed several control experiments to test whether the IFN γ YFP reporter faithfully identified the TCR-dependent expression of this critical effector cytokine. It should be noted that this reporter is built so that YFP is expressed from the same transcript as endogenous IFN γ , so YFP expression must be linked directly to IFN γ expression and we demonstrate this direct correlation in Supplemental Figure 1. We also demonstrate that only effector CD8 $^+$ T cells in the skin become YFP $^+$ and this requires cognate antigen recognition. We also show that all CD8 $^+$ T cells isolated from the skin will become YFP $^+$ following activation with peptide. Finally, we show that many genes known to be targets of the TCR signaling pathway are more highly expressed in YFP $^+$ T cells (IFN γ , Ccl3, Ccl4, perforin, ICOS, CD69, and PD-1). Of course, there is likely no such thing as a “100% perfect” gene expression reporter system, but clearly our approach was successful in identifying the transcriptional consequences of antigen recognition within the skin microenvironment, which allowed us to proceed with mechanistic studies about how antigen recognition by effector CD8 $^+$ T cells influences tissue-residency programming.

Minor points:

The correct nomenclature of the mouse MHC is H-2D instead of H2-D.

We have fixed this typo throughout the manuscript.

I talked to dermatologists, but the term parenchyma is not normally used in the context of skin. The term should therefore be replaced.

We have removed this term throughout the manuscript.

In the introduction, the authors make several statements that should be changed. Line 43: T cells are not activated only in draining lymph nodes (e.g., spleen, tertiary lymphoid structures...). Line 44-46: It is not proven that inflammation must subside before T_{rm} are formed. Line 64: The authors talk about the main regulator of T_{rm}, but did not define what they consider as the main regulator.

We have made modifications to the introduction of our manuscript.

In general, the introduction is long and it takes quite a while to become clear what issues have been addressed.

We have made changes to the introduction accordingly.

It is difficult to understand what the authors are trying to say in the sentences in lines 98 to 103.

We have modified this statement accordingly.

Reviewer #3 (Remarks to the Author):

The molecular mechanisms regulating the formation and maintenance of tissue-resident memory T cells (Trm) are still incompletely understood. Specifically, it is not known how a second encounter with antigen in the tissue after the initial priming in lymph node can influence T cell tissue residence. The Nolz lab has previously shown that antigen in skin facilitates tissue-resident memory T cell formation during vaccinia infection. In this follow-up paper, Abdelbary et al. presents further evidence showing the critical role for tissue antigen in facilitating Trm formation. Critically, the authors demonstrate multiple mechanisms by which tissue antigen enable Trm formation: re-encounter with antigen upregulates Blimp1 transcription factor, which is required for changes in T cell chemotactic properties to promote tissue residency and for Trm residency. The authors also suggest that T cell receptor signal strength regulates T cell chemotactic properties, with low affinity antigen being sufficient to induce migration towards CXCR6, but with strong antigens required for suppression of S1P egress. These are important findings in rapidly developing field in immunology. The experiments are very well designed, and the manuscript is clearly written.

However, there are few questions/concerns that should be address by the authors:

1. Use of IFN γ -YFP reporter: The authors present very convincing data that % of YFP+ antigen-specific T cells correlates with % of IFN-g+ cells. However, this is only shown for T cells from spleen after in vitro peptide re-stimulation. Are YFP+ T cells from skin also IFN γ +, both ex vivo and with or without antigen re-stimulation? How long can the YFP signal persist after cessation of antigen encounter? This is important for interpretation of the IFN γ -YFP model.

To estimate the half-life of YFP, we stimulated effector P14 CD8⁺ T cells from the spleen overnight with GP₃₃₋₄₁ peptide, then washed the T cells extensively and monitored loss of YFP expression over time. These data are now shown in Figure S1C and suggest that the half-life of YFP using our experimental approach is approximately 14.2 hours. It should also be noted that this reporter is built so that YFP is expressed from the same transcript as endogenous IFN γ , so YFP expression must be linked directly to IFN γ gene expression and we demonstrate this direct correlation in Supplemental Figure 1 (as pointed out by the reviewer). We would also like to highlight that in our gene expression analysis, we found that IFN γ expression was significantly higher in the YFP+ T cells than the YFP- T cells and that loss of YFP expression in the skin occurs rapidly as viral infection is cleared (this is now highlighted in Figure 1B). Finally, we analyzed expression of IFN γ protein directly ex vivo and we could only detect IFN γ expression in the YFP+ T cells from the skin and not in the YFP- T cells from either the skin or spleen. The IFN γ protein signal is not as “robust” as the YFP signal, and we hope the reviewer appreciates that identifying IFN γ expression within T cells directly ex vivo is not trivial and likely underrepresents expression of this cytokine in vivo. This was the rationale for using the IFN γ -YFP reporter system for this study in the first place.

2. Figure 1H and I. Only KLRG1^{low} cells express in this experimental system. How does this relate to previous work, showing that both KLRG1^{low} and KLRG1^{hi} T cells can produce IFN-gamma (for example: Robbins et al. (2003) Differential Regulation of Killer Cell Lectin-Like Receptor G1 Expression on T Cells). Are KLRG1^{low} T cells from skin also CD127^{hi}? Can KLRG1^{low} cells from skin produce

IFN-gamma after antigen re-stimulation in vitro? Does IFN-gamma production differ between KLRG1^{low} and KLRG1^{hi} populations from skin and from spleen?

In response to questions from both R1 and R3 regarding IFN γ expression compared to KLRG1 expression, we revisited this finding for clarity. When comparing skin +/- antigen, we found that KLRG1^{Lo} T cells were found preferentially in the skin regardless of the presence of cognate antigen. Thus, this finding likely reflects selective migration of KLRG1^{Lo} T cells into the skin and KLRG1^{Hi} T cells remain more associated with the vascular (as suggested by R1). We have now included these new data in Figure 1L-O. We also did not intend to imply in our narrative that KLRG1^{Hi} T cells were not able to produce IFN γ , as all T cells from the skin and spleen are able to produce IFN γ following ex vivo peptide stimulation as shown in Supplemental Figure 1H,I.

We have previously reported that KLRG1^{Lo} CD8⁺ T cells in the skin also express CD127 (Khan et al, JEM, 2016).

3. Figure 1N. What is the time point for the analysis? The figure legend suggests it is day 7, but this does not agree with the conclusion on T_{rm} formation after viral clearance.

We apologize for this confusion within the figure legend. The data in the original Figure 1N was day 40 after infection (which we did not clearly indicate). Based also on a similar question/concern of reviewer 1, we have updated this figure and legend accordingly (now Figure 1P-R).

4. Figure 5J-M: What is the time point for analysis of Blimp1 upregulation? Naïve T cells have been reported to express Blimp1 after 48h of stimulation (Gong and Malek, 2007 Cytokine-Dependent Blimp-1 Expression in Activated T Cells Inhibits IL-2 Production), so the difference observed by the authors could be due to the different kinetics of Blimp1 expression between naïve and pre-activated T cells.

For Figure 5J-M, Blimp1 expression was analyzed 24 hours after peptide stimulation. We agree with the reviewer that this difference in expression is likely due to differences between naïve and previously activated effector CD8⁺ T cells. In fact, Martins et al, Nature Immunology, 2006 reported that expression of Blimp1 by naïve T cells took 6 days of in vitro stimulation. Thus, we believe these data collectively show that effector CD8⁺ T cells become poised to rapidly express Blimp1 following secondary antigen encounter that, in turn, is necessary to regulate the chemotactic properties of effector CD8⁺ T cells (as shown in Figure 7). We have modified this figure legend accordingly.

5. Blimp1 conditional deletion – in the main text, the Blimp1^{-/-} mice are only introduced as Prdm1^{fl/fl}. These must have been Cre-ERT2⁺, as this Cre is mentioned in the Material and Methods, and mice are treated with tamoxifen. Please mention the Cre in the main text. What exactly were the Blimp1 WT controls used in these experiments – were they also Cre⁺ but Prdmwt/wt; where they also tamoxifen-treated? Please add data confirming the efficiency of deletion after tamoxifen treatment.

The experimental design for our results shown in Figures 6/7 is described in Figure 6A. Prdm1^{+/-} and Prdm1^{fl/fl} P14 CD8⁺ T cells both expressed ROSA26-Cre-ERT2. These T cells were mixed and transferred into the same animals that were then treated with tamoxifen for 5 days. As requested, we have made additions to the text to clearly identify the T cells used for the experiments.

Data confirming the efficiency of Blimp1 deletion is now provided in Fig S6.

6. P14 and OTI T cells used for the adoptive transfer experiments – were the T cells sorted for naïve (CD44^{low}, CD62L^{hi}) phenotype? If not, what was the % of the naïve phenotype cells used for adoptive transfers.

TCR-tg T cells from mice in our colony are consistently >95% naïve (CD44^{Lo}, CD62L^{Hi}) prior to adoptive transfer. We clarified this point in the methods section under “Mice and Infections”

7. Information on number of independent experiments using biological/technical replicates should be included in the figure legends. The means/individual data points are from single representative experiment, or from data pooled from multiple experiments?

We have made additions to the figure legends to address these questions.

8. In some cases (for example: 2F-1), t-tests seem to be used compare more than 2 means. Please either change this to more appropriate analysis or explain how the data was analysed.

We have modified the figure legends accordingly. The Statistical significance (for Figure 2G,H) was calculated using a one-way ANOVA followed by Dunnett’s multiple comparisons test using VacV-N4 as the control group.

REVIEWERS' COMMENTS

Reviewer #1 (Remarks to the Author):

The paper is improved by the revisions. With the suggested clarifications, the study is informative and an important contribution to the literature.

Reviewer #2 (Remarks to the Author):

I have to admit that the authors made only limited efforts in responding to the points I raised. To me, the central finding of the study is that TCR signal strength in tissues (and not just the degree of stimulation in SLO / lymph nodes) affects T_{rm} formation, which I think is interesting and worth noting. I still don't think the abstract conveys the message clearly. I also find it difficult to understand why the authors do not agree with this conclusion in the reply brief when they essentially confirm it in their response.

The authors chose to emphasize the importance of the Blimp-CXCR6 axis, which is not an unknown mechanism for T_{rm} formation (ie, PMIDs 35389518, 30692620).

The authors conclude in the abstract "Collectively, our findings show that access to antigen presentation and strength of TCR-signaling required for Blimp1 expression establishes the chemotactic properties". I had asked for mechanistic proof of this point, which the authors did not provide in the revision. Currently, this point remains solely at the correlative level, i.e., higher signal strength more Blimp, but whether differential Blimp activity is indeed the mechanism driving differential numbers of T_{rm} following high or low affinity stimulation in tissues remains unresolved.

The author's main conclusion therefore remains unsubstantiated, raising significant concerns about the publication of the manuscript in its present form.

REVIEWERS' COMMENTS

Reviewer #1 (Remarks to the Author):

The paper is improved by the revisions. With the suggested clarifications, the study is informative and an important contribution to the literature.

We thank the reviewer for the constructive feedback that led to improved clarity of the manuscript.

Reviewer #2 (Remarks to the Author):

I have to admit that the authors made only limited affords in responding to the points I raised. To me, the central finding of the study is that TCR signal strength in tissues (and not just the degree of stimulation in SLO / lymph nodes) affects Trm formation, which I think is interesting and worth noting. I still don't think the abstract conveys the message clearly. I also find it difficult to understand why the authors do not agree with this conclusion in the reply brief when they essentially confirm it in their response.

We have modified the abstract during this revision to try to convey this message more clearly, as requested by the reviewer.

The authors chose to emphasize the importance of the Blimp-CXCR6 axis, which is not an unknown mechanism for Trm formation (ie, PMIDs 35389518, 30692620).

The authors conclude in the abstract "Collectively, our findings show that access to antigen presentation and strength of TCR-signaling required for Blimp1 expression establishes the chemotactic properties ". I had asked for mechanistic proof of this point, which the authors did not provide in the revision. Currently, this point remains solely at the correlative level, i.e., higher signal strength more Blimp, but whether differential Blimp activity is indeed the mechanism driving differential numbers of Trm following high or low affinity stimulation in tissues remains unresolved.

The author's main conclusion therefore remains unsubstantiated, raising significant concerns about the publication of the manuscript in its present form.

We strongly disagree that our statement "Collectively, our findings show that access to antigen presentation and strength of TCR-signaling required for Blimp1 expression establishes the chemotactic properties" is only a correlation and that our conclusions are unsubstantiated. In fact, we have shown the following:

In Figure 5, we characterized extensively how different levels of TCR signaling leads to changes in Blimp1 expression. We clearly show that strong TCR signaling was required for Blimp1 expression and to prevent S1P-mediated migration. Surprisingly, we found that even weak TCR agonists caused CXCR6 to become expressed. Based on those findings (as well as our gene expression analysis), we then proceeded to determine whether Blimp1 was required for tissue-resident T cells to form and whether this required cognate antigen-recognition. Clearly, our extensive analysis performed in Figure 6 demonstrate that BOTH antigen recognition and Blimp1 expression are necessary for tissue-resident T cells to form in the skin during viral infection. Thus, we demonstrate that even in response to a strong TCR agonist (GP33) that tissue-resident T cells do not form without Blimp1, so it is unclear why the reviewer suggests that TCR signal strength necessary for Blimp1 expression (that we demonstrate in Figure 5) and the absolute requirement for Blimp1 for tissue-residency differentiation to occur is simply a correlation. We then provide even more in-depth mechanistic analysis in Figure 7. Here, we show that without Blimp1, expression of CD69 (as an indirect readout for downregulation of S1PR1) and CXCR6 are both significantly reduced compared to WT T cells responding to a strong TCR agonist (GP33). In fact, expression of these critical genes closely resembles their expression in skin without any cognate antigen (see Figure 4). Nevertheless, we went on to perform migration assays to definitely prove that Blimp1 is necessary for altering chemotaxis (both CXCL16 and S1P) in response to a strong TCR agonist. Overall, these data presented in the manuscript clearly show that different affinity TCR agonists lead to differential Blimp1 expression and that Blimp1 is required for both changes in chemotactic properties and for tissue-resident T cells to form, thus, extensive scientific evidence that completely substantiates our overall conclusion of our study.

We have added the following sentence into the DISCUSSION (Page 22): "Although our study here focused primarily on Blimp1-dependent changes in T cell migration, it will be of interest to also investigate whether other signaling pathways that are regulated by strength of antigen stimulation (differential expression of ICOS, for example) could also contribute to T_{RM} differentiation within non-lymphoid tissues such as the skin."